

# Variability in a four-network composite of atmospheric CO₂ differences between three primary baseline sites

Roger J. Francey, Jorgen S. Frederiksen, L. Paul Steele and Ray L. Langenfelds

CSIRO Oceans and Atmosphere, Aspendale, Victoria, AUSTRALIA
5    Correspondence to: Roger J. Francey (roger.francey@csiro.au)

**Abstract.** Spatial differences in the monthly baseline $CO_2$ since 1992 from Mauna Loa, (mlo, 19.5°N, 155.6°W, 3379m), Cape Grim (cgo, 40.7°S, 144.7°E, 94m) and South Pole (spo, 90°S, 2810m), are examined for consistency between four monitoring networks. For each site pair, a composite based on the average of NOAA, 10    CSIRO and two independent SIO analysis methods is presented. Averages of the monthly standard deviations are 0.25, 0.23 and 0.16 ppm for mlo-cgo, mlo-spo and cgo-spo respectively. This high degree of consistency and near-monthly temporal differentiation (compared to $CO_2$ growth rates) provides an opportunity to use the composite differences for verification of global carbon cycle model simulations.

Interhemispheric $CO_2$ variation is predominantly imparted by the mlo data. The peaks and dips of the seasonal 15    variation in interhemispheric difference act largely independently. The peaks mainly occur in May, near the peak of Northern Hemisphere terrestrial respiration. Boreal spring is when interhemispheric exchange via eddy processes dominates, with increasing contributions from mean transport into boreal summer. The dips occur in September, when the $CO_2$ partial pressure difference is near zero, just after the peak in the mean interhemispheric exchange via the Hadley circulation. Surface-air terrestrial flux anomalies would need to be up 20    to an order of magnitude larger than found in order to explain the peak and dip $CO_2$ variations (large enough to significantly influence short-term northern hemisphere growth rate variations).

Recent features in the composite records, inconsistent in timing and amplitude with air-surface fluxes, are largely consistent with interhemispheric transport variations. These include the remarkable stability in annual $CO_2$ inter-hemispheric difference in the 5-year relatively ENSO-quiet period 2010-2014, and the 2017 recovery 25    in the $CO_2$ interhemispheric gradient from the unprecedented El Niño event in 2015-16.



## 1 Introduction

Atmospheric $CO_2$ measurements are normally introduced into global carbon budgets as a "global growth rate ...
based on the average of multiple stations selected from the marine boundary layer sites with well mixed background air ..., after fitting each station with a smoothed curve as a function of time, and averaging by latitude band …" (Le Quéré et al., 2018).

Interhemispheric (IH) baseline $CO_2$ differences have been linked to intermittency in seasonal exchange between hemispheres by Francey and Frederiksen, 2016 (FF16) and Frederiksen and Francey, 2018 (FF18). The
dynamical processes described in these studies, if not adequately captured in global carbon cycle model transport, will compromise conventional approaches that assume sub-annual $CO_2$ variation is primarily the result of terrestrial exchange (e.g. Yue et al., 2017, Rödenbeck et al., 2018).

FF16 and FF18 focussed on Commonwealth Scientific and Industrial Research Organisation (CSIRO) $CO_2$ data. However, co-sampling over 25 years of background $CO_2$ has occurred at three iconic remote sites by four well-
established global monitoring flask sampling networks, two from Scripps Institution of Oceanography (SIO1, SIO2), one from the National Oceanic and Atmospheric Administration (NOAA) and one from CSIRO. Consistency in a composite of within-network spatial differences provide an improved opportunity to examine inherent assumptions in the statistical treatment of atmospheric data in global carbon budgets.

Compared to $CO_2$ information used in typical growth rate studies, the 25-year composites of within-network
spatial differences measured by the flask networks provide additional insights, summarised here:

- The effective time resolution can be short, circumventing ambiguities between seasonality and inter-annual variability, and 6-9 month end-effects inherent in conventional growth rate analyses. The latter are generally obtained by harmonic filtering to define seasonality, and 22 month smoothing to define inter-annual behaviour (e.g. Thoning et al., 1989). The monthly average spatial differences used here
have higher resolution, being obtained from 80-day smoothed records that represent the available flask measurements.

- Since flask samples from each network are analysed in a central laboratory, bias associated with calibration of an internal calibration scale relative to the international $CO_2$ WMO mole fraction scale (Zhou and Tans, 2006), and similar bias when relating a reference gas to the internal scale, generally
cancel for within-network site differences.

- The IH $CO_2$ differences suppress the influence of equatorial surface exchanges that are uplifted to an altitude where they can mix into both hemispheres. This improves sensitivity to cross-equatorial atmospheric fluxes that occur in a region where transport is less well-defined than at higher latitudes (e.g. Lintner et al., 2004).

- Multi-species (e.g. other long-lived trace gases and their isotopes) with different biogeochemistry but identical time of collection are often available (particularly from pressurized flasks). A multi-species approach was briefly assessed in FF16 and FF18 and is the subject of further studies.

Disadvantages of flask compared to *in situ* $CO_2$ sampling include:

- $CO_2$ artefacts related to extended storage of air in flasks, mainly affecting sampling from remote sites
with annual resupply (in this study spo). Flask size, permeation through elastomer seals in pressurized flasks (Sturm et al., 2004), and inadequate surface preconditioning to limit $CO_2$ adsorption on surfaces, are possible contributing factors. In this study, factors such as the use of larger and/or unpressurized



flasks at spo, and corrections informed by co-sampling with *in situ* analysers (e.g. Stavert et al., 2019), help address this concern.

• The brevity of sampling compared to *in situ* measurement. Monthly average concentrations from each site comprise the average of 1-10 or more flask samples (depending on network and site) with the filling of each flask taking approximately 1-20 minutes (depending on differing sampling strategies to precondition and/or pressurize). That is, air sampled over a few hours can be used to represent a monthly value. The effectiveness of baseline selection becomes a critical issue. In the current study it will be seen

that the generally small standard deviations in monthly averages across networks with quite different sampling frequency imply that this is unlikely to be a major concern with the selected primary baseline site monthly data employed.

Both types of measurement, but particularly flasks at remote sites because of the delays between sample collection and analysis, are susceptible to leaks/contamination in sample intake lines. Inevitable variations in

quality at any one site or laboratory are de-emphasized in the composite averages (but remain reflected in ensemble standard deviations). There is a range of sampling and measurement strategies across networks that provides a robustness when consistencies persist between networks, but it can complicate the attribution of inconsistencies.

The data used here are monthly averages obtained from ftp sites at NOAA, SIO and CSIRO (listed below under

Data Availability), and are also available from the World Data Centre for Greenhouse Gases (https://gaw.kishou.go.jp/search). No editing or selecting of pre-existing web-sourced data has occurred, since there is sufficient data that periods of consistency dominate statistical comparisons.

## 2 Background information on flask networks

By 1958 C.D. Keeling had identified mlo and spo as optimum sites to obtain background $CO_2$ in the respective

hemispheres and by the 1970s was obtaining a regular monthly supply of air admitted to 5L evacuated glass flasks from both sites (SIO1: Keeling C. D. et al., 2001). Since 1992, there are $CO_2$ measurements as a by-product of a global network focussed on $O_2/N_2$ ratios in baseline air (SIO2: Keeling R. F. and Schertz, 1992); this program uses 5L glass flasks, with air cryogenically dried, flushed and filled to ambient pressure. While there is commonality with regard to calibration, in the context of spatial differences the SIO networks can be

considered independent.

NOAA began sampling from all three sites (as part of a much larger network) from 1984, using a variety of flask and filling methods. From around 1992 the current system of Peltier-dried air in pressurized 2.5 L flasks (Tans et al., 1992, Conway et al., 1994, Dlugokencky et al., 2014) was phased in. They have maintained the WMO Central $CO_2$ Calibration Laboratory since 1996 (a role previously carried out by SIO). The NOAA

atmospheric sampling is generally more frequent (typically 8-10 flasks per month) than is the case for SIO or CSIRO programs (except for the CSIRO cgo program). However, the size and sampling frequency in the NOAA network amplifies calibration challenges (e.g. due to shorter lifetimes of reference and calibration standards). Both NOAA and SIO use non-dispersive infra-red analysers (NDIR) for $CO_2$ measurement.

CSIRO flask sampling at cgo, spo and mlo in the early 1980s used chemically dried air, pressurized into 5L

glass flasks, using NDIR for analysis; however, analyses here are restricted to CSIRO's more comprehensive measurements from 1992 using chemically-dried, pressurized air in 0.5L glass flasks, but with retention of 5L flasks at spo (Francey et al., 1996). In the early 1990s the use of gas chromatography with flame ionisation



detection (GC/FID) was exploited to monitor $CO_2$ in flasks at CSIRO. The GC/FID technique used provides a significantly more linear response for $CO_2$ than NDIR (see Supplement Figure S1) and has required much

smaller samples than most other equipment employed consistently over the 25 years; both factors contribute to calibration integrity. Counterbalancing the fact that smaller sample flasks imply enhanced susceptibility to flask storage effects, the small sample requirement has permitted pioneering, long-term "same-air" measurement inter-comparisons between NOAA and CSIRO on NOAA cgo samples (Masarie et al., 2001; see Supplement Figure S3). Hourly radon measurements at Cape Grim (Chambers et al. 2016) were also introduced around this

time. Cape Grim sampling is further informed by a decade of vertical profiling (Langenfelds, 2003; Pak, 1996), back trajectory analysis, and other tracers (e.g. Dunse et al., 2001), demonstrating that selected cgo data can achieve a degree of spatial representativeness that matches, or sometime exceeds, that at the more remote high-altitude sites mlo and spo.

Note: The challenges of maintaining high quality over 25 years in any one monitoring program are many. They

include external factors, acknowledged but not pursed here, such as high turnover of skilled staff particularly at remote monitoring sites or changes in institutional strategic and economic priorities. The latter are well described by Keeling (1998), with CSIRO sharing similar institutional experiences.

Before combining data from different networks, systematic differences between programs and sites are examined in Figure 1. It shows monthly differences from NOAA data of SIO1, SIO2 and CSIRO data, for each

of the three baseline sites. Five-month running means aid discussion. NOAA, because it has the most extensive global network, and since 1996 has also operated the WMO $CO_2$ Central Calibration laboratory, is selected as the reference. The $CO_2$ mixing ratios used here are referred to in the commonly used units of parts per million (ppm) rather than the more strictly correct term of μmole of $CO_2$ per mole of dry air.

In Figure 1, there is clear evidence of systematic difference in mean offsets, seasonality, and between sites

within one network. In the context of inter-hemispheric exchange, the typical 0.5 ppm range of variation remains relatively small compared to the 7-10 ppm maximum IH $CO_2$ (when most exchange occurs). Also, when calculating within-network IH differences consistent calibration offsets at all three sites will largely cancel.

The means and bracketed standard deviations in the differences from NOAA that are included in Figure 1 are

calculated for data from the years 1996-2016; prior to 1996, variability is generally greater and corresponds to a period of change and development in all three laboratories. For example, there is a marked inconsistency between NOAA mlo-spo and mlo-cgo in 1991-1993, particularly in seasonal amplitude; CSIRO has comparable measurements that are more consistent (see Supplement Figure S3). This is a further reason for caution with the data in this period.

In the post-1996 statistics the SIO1 offsets and scatter from NOAA behave similarly for mlo and spo. This is not the case for SIO2, which has similar offsets at mlo (-0.18 ppm) and cgo (-0.19 ppm), but not at spo (-0.04 ppm), or for CSIRO (mlo: -0.08, cgo: +0.01, spo: +0.13 ppm). SIO2 mlo data exhibit the largest scatter (±0.37 ppm). The CSIRO records at cgo exhibit the smallest offset and scatter relative to NOAA (+0.01 ± 0.08 ppm), but remnant seasonality is still evident. A possible consideration here is that the CSIRO GC/FID near-linear

response for $CO_2$ means that results are not sensitive to differences between sample and reference $CO_2$, a factor reinforced in the SH if reference gases use recent SH baseline air. This is generally not the case for non-linear NDIR measurement and particularly in the NH if relatively short-lived reference gases sourced in the NH do not match ambient $CO_2$ from a site.



Systematic differences due to sampling and measurement methodology are likely the result of a combination of
the linearity of instrument response and flask storage effects (particularly at spo and particularly in CSIRO spo
data with the sparsest sample density), that is likely to be partly addressed with a comprehensive review of
metadata. This is outside the scope of this study so we rely on the ensemble of average differences to moderate
their influence.

Figure 2 provides an overview of the impact of measurement bias on spatial differences. Data are presented as
3-month seasonal averages in order to minimize potential influences related to network sample-frequency (by
ensuring an adequate number of individual flask samples per season). As well, the seasonal selection
distinguishes periods of relatively stable partial pressure differences between hemispheres and the selected
seasons also distinguish eddy and mean IH transport mechanisms (FF18).

- For the most part, and particularly in the Aug-Oct season when IH $CO_2$ difference is at a minimum, there
is a high level of consistency in the year-to-year variation in seasonal spatial differences from each
network.

- There are relatively few examples of one record differing markedly from the others; when it occurs it is
often for reasons evident in Figure 1. For example in Figure 2 NOAA cgo-spo appears low in 1992/1993;
CSIRO mlo-spo shows negative outliers in May-Jul 2009 and Nov-Jan 2002, but not at mlo-cgo. SIO2
outliers in 2002 and 2006 exhibit similar characteristics; positive outliers e.g. SIO2 Nov-Jan 2016
suggests a cgo problem. In Feb-Apr 2005 NOAA data indicate a possible mlo problem; however this is
also when the 'volatility' of the records (and in IH transport) is large, so it is conceivable that different
flask sampling numbers and times could contribute to lower values by both SIO and CSIRO. Closer
inspection of individual flask metadata may resolve these infrequent anomalies, but for the present,
composite averaging is relied on to moderate their influence.

- The largest IH $CO_2$ variability is recorded in Feb-Apr and in May-Jul, both seasons having near-equally
large IH differences. These seasons correspond to maximum respiration from NH forests. Feb-Apr is also
when IH exchange by eddy processes is most influential (FF16), whereas mean transport via the Hadley
circulation is the main dynamical influence in May-Jul (FF18).

FF18 provided correlations of mlo-cgo with wind indices representing both eddy and mean IH transport (using
CSIRO data only), emphasizing temporal covariation. FF16 examined the relationship between eddy transport
indices and IH $CO_2$ differences for CSIRO mlo-cgo and SIO1 mlo-spo data. Here we individually examine
major anomalies in the 25-year composite records with more emphasis on the magnitude and timing of
influences that might be attributable to IH transport variation.

**3 Composite records of baseline station spatial differences**

For each of mlo-cgo, mlo-spo and cgo-spo monthly $CO_2$ differences, Table 1 shows the number of months
between 1992 and 2016 contributing to a composite value, arranged in columns indicating the number of
contributing networks; e.g. 256 of 300 months have 4 networks contributing to mlo-spo, while 277/300 months
have three contributing networks at mlo-cgo.

The percentage of missing months for each network, and scatter in the composite differences for different
historic periods is tabulated in the Supplement Tables S1, S2. Data analysed below extend to 2017.

The monthly composite $CO_2$ differences are shown in Figure 3 (and tabulated in our Supplement Table S3).
Error bars represent the ensemble standard deviation (except for the cgo-spo case, Feb 2009, with only the



NOAA network contributing; it is arbitrarily assigned 100% uncertainty and appears as an outlier in Figure 3c).

The seasonality at mlo, generally attributed to NH forest photosynthesis/respiration cycles, is the dominant variation in the IH $CO_2$ differences. The composite errors are small by comparison. Standard deviations of mlo-cgo, mlo-spo and cgo-spo are 0.25, 0.23 and 0.16 ppm respectively. Systematic inter-annual variability is well-defined and is reflected similarly in both IH records, and is consistent with mlo driving most of the variation. Variations that exceed the ensemble monthly standard deviations include:

•    The overall increase in IH difference, generally attributed mainly to increasing NH fossil fuel $CO_2$ emissions is indicated by a linear regression through the mlo-cgo values (with slope 0.061 ppm $yr^{-1}$; mlo-spo gives 0.060 ppm $yr^{-1}$). The slope of regressions is much higher for the Apr-May data (0.09 ppm $yr^{-1}$) than for Sep-Oct data (0.05 ppm $yr^{-1}$), slightly less than the overall mean.

   •    From 1992-2017, the majority of minima occur in Sep; of 26 minima, 24 occur in Sep and 2 in Oct
(1992 & 1995). Of the 26 maxima 20 are in May, and 6 in Apr (1997, 1999, 2000, 2004, 2005, 2016).

   •    Scatter in the amplitude of seasonal maxima (boreal winter/spring) is smaller before 1999. The step-like behaviour in Apr-May from 2009 to 2010 remains the major anomaly.

   •    In contrast, the minima (in boreal summer/autumn) exhibit greater scatter before 2011, replaced afterwards by a smooth decline to a marked 2016 minimum, then sudden reset in 2017.

•    Unusually low boreal summer/autumn IH minima also occur in 1993-1994. Apart from being a period when measurement and calibration methods were consolidating (as discussed next section) the most significant volcanic influence (Pinatubo) is potentially an influence at this time.

This raises a question as to how well mlo data represents the Northern Hemisphere. Of more relevance to this study is how well do the mlo samples represent air that is transferred into the Southern Hemisphere. Flasks are
collected at mlo above 3 km altitude in down-slope winds, close to the upper troposphere regions where the IH transfer processes defined in FF18 occur (see Figure 5), circumstances not shared by other NH surface monitoring sites.

Unlike in typical growth rate analyses, the peak and trough values are largely independent. This is visually explored in Figure 3 using spline polylines linking peaks (solid) and dips (dashed) months of IH $CO_2$
differences. Typical trace gas mixing within extra-tropical hemispheres is typically estimated at 1-2 months or less, and inter-hemispheric exchange times estimated at 6-12 months or more (e.g. Bowman and Cowan, 1997, Jacob, 1999). Monthly changes in the peak and trough IH $CO_2$ largely reflect flux changes in or out of the extra-tropical northern troposphere close to that month. The following sections seek similarities with possible causal forcing processes.

**4 Processes influencing $CO_2$ IH difference variations**

Variations in atmospheric $CO_2$ global carbon cycle models generally attribute short term variations in the global budget to exchanges with the terrestrial biosphere (Le Quéré et al., 2018; Yue et al., 2017) and implicitly assume model atmospheric transport is correct. While the models have demonstrated an impressive ability to predict mid-to-high latitude $CO_2$ variations influenced by weather, it is less clear that short term variations in IH
exchange have been adequately captured.

To exploit the sharp definition of the peaks and dips of IH $CO_2$ in Figure 3, their amplitude and timing are initially compared to anomalies in monthly estimates of terrestrial biosphere emissions and wildfires in Figure 4.





It is assumed here that flux variations from ocean sources are much smaller than terrestrial and IH atmospheric flux variations and can be generally neglected.

**4.1 Terrestrial fluxes influencing IH Transport:**

The primary determinant of the well-defined seasonality in IH $CO_2$ in Figure 3 is the temperature-moderated photosynthesis/respiration cycle of NH forests. The influence of equatorial vegetation surface-air $CO_2$ exchange that mixes into both hemispheres is suppressed in IH $CO_2$. The issue here is whether anomalies in the IH $CO_2$ are related to anomalies in terrestrial exchange.

Monthly Dynamic Vegetation Model (DVM) estimates of Net Terrestrial Biosphere Production (NBP) in three latitude bands 90ºN-30ºN, 30ºN to 30ºS and 30ºS-90ºS over the 1992-2016 period with the Community Atmosphere Biosphere Land Exchange (CABLE) model (Kowalczyk et al., 2006; Haverd et al., 2018) are used. NBP signs are reversed and are described as terrestrial-to-air carbon fluxes. Global wildfire emissions from the Global Fire Emissions Database (Randerson et al., 2018, GFED4.1) from 1997-2015 are classified as NH, EQ

and EQ/SH, and are also examined. For each data set, anomalies in seasonal behaviour for each latitude band were determined by subtracting the mean seasonality from the monthly value.

The major seasonal anomalies in NBP and Wildfire emissions that potentially influence IH $CO_2$ are shown in Figures 4(a) and 4(b). The largest anomalous surface to air flux is the extreme equatorial emission anomaly from equatorial wildfire in late 1997 (~0.9 PgC over 3 months) and is not associated with unusual behaviour in

the IH $CO_2$ records.

With rapid mixing within the extra tropical northern hemisphere, as rapidly as 1-2 weeks (Jacob, 1999), and since IH $CO_2$ peaks re-occurring at the same time within 1 month of each year, if terrestrial NH emissions were a major determinant, close correspondence in timing of terrestrial anomalies and the IH $CO_2$ peaks would be expected. This is not evident in Figure 4. More importantly, the amplitude of terrestrial anomalies appear to be

far too small to influence the IH $CO_2$ peaks and dips changes.

Over the last 25 years the annual relationship between global (mainly NH) fossil fuel combustion emissions and IH difference has been 2.8 PgC ppm$^{-1}$ (0.36 ppm (PgC)$^{-1}$, FF18, see also Figure 9 below). This is when northern fossil fuel emissions effectively mix globally. The volume of the troposphere north of Mauna Loa is around 33% of the global troposphere, so that on the shorter time frame of within-hemisphere mixing, only ~0.92 PgC

is required to change the NH background $CO_2$ by 1 ppm. In Figure 4(c) we round this to 1 PgC = 1 ppm for simplicity.

The IH $CO_2$ fluxes necessary to explain variations in the peaks (Apr-May) and dips (Sep-Oct) remain almost an order of magnitude greater than the anomalies in estimated terrestrial biosphere and wildfire fluxes.

Accepting the precision and near-hemispheric spatial representation of the composite IH $CO_2$ records, these

inconsistences with terrestrial emissions in both timing and magnitude suggest that there are other short-term influences on IH $CO_2$ of greater magnitude than terrestrial exchange.

**4.2 Wind indices reflecting IH Transport:**

In contrast to the case for air-surface exchanges, there are a number of prominent features in the composite IH $CO_2$ records that are shared with behaviour in the dynamical indices of FF18. Inter-hemispheric exchange of

$CO_2$ occurs mainly by eddy processes in the boreal winter-spring and by mean convection and advection associated with the Hadley circulation in the boreal summer-autumn (FF18 and references therein). FF18





developed wind indices that characterize both types of IH transport based on reanalysis data sets focusing on the National Center for Environmental Prediction (NCEP) and National Center for Atmospheric Research (NCAR) reanalysis (NRR) data (Kalnay et al., 1996). Eddy transport is described by $u_{duct}$, the average 300 hPa zonal

velocity in the Pacific Westerly duct region (Frederiksen and Webster, 1988) of 5ºN to 5ºS, 140 to 170ºW (FF16, FF18). Here we use that index and two of the four indices for mean transport introduced in FF18. These are $\omega_P$, the average 300 hPa vertical velocity in pressure coordinates in the region 10-15ºN, 120 to 240ºE, and $v_P$ the average 200 hPa meridional velocity in the region 5-10ºN, 120 to 240ºE. Figure 5 provides a schematic of the geographical location of regions used by FF18, and time series of the monthly values of wind indices are

shown in Figure 6.

The top panel in Figure 6a shows the $u_{duct}$ index which characterizes cross-equatorial Rossby wave dispersion, Rossby wave breaking and corresponding increases in transient kinetic energy and eddy transport in the near-equatorial upper troposphere (Webster and Holton, 1982, Frederiksen and Webster, 1988, Ortega et al., 2018). The large scale Rossby waves are generated by thermal anomalies and topographic features including the

Himalayan mountains from which they propagate south-eastward and are able to penetrate into the SH when $u_{duct}$ is positive, corresponding to an open Pacific Westerly duct.

The $\omega_P$ and $v_P$ indices in Figures 6b and 6c describe the strength of the mean transport by the Hadley cell in the Pacific region with negative $\omega_P$ corresponding to uplift and negative $v_P$ to north to south transport.

Net interhemispheric trace gas exchange requires a partial pressure difference between hemispheres. For $CO_2$

the average seasonal cycle of 25-year mean partial pressure difference, represented here by monthly baseline mlo-cgo, is shown in Figure 7(a) (mlo-spo is not shown since it is effectively identical).

The positive mean IH $CO_2$ is largely due to fossil fuel emissions. Months of positive (north-south) IH difference are shaded green and only in Sep-Oct is there a small reverse gradient. Transport of $CO_2$ from the northern to the southern hemisphere occurs when green shaded areas in Fig. 7(a) coincide (on average) with blue shaded

areas (Fig. 7(b), via eddy transfer with index $u_{duct}$), or with red shaded areas (Figs. 7(c) and 7(d), via mean transport with indices $\omega_P$ and $v_P$).

Figure 7 also demonstrates that differences from the long-term mean in transport indices (average for each month) vary between the significant El Niño events in 1998, 2010 and 2016:

- In 2010, the IH $CO_2$ difference exceeds the average between Feb-Jul (Fig. 7(a)) with reduced eddy
transfer between Feb-Apr, associated with lower that average $u_{duct}$ (Fig. 7(b)). Further, between Jun-Sep, there is weaker ascent (Fig. 7(c)) and north to south upper tropospheric wind (Fig. 7(d)) in the key regions defining $\omega_P$ and $v_P$. As noted in FF18, the IH $CO_2$ eddy and mean transports reinforce to contribute to the unprecedented 2009 to 2010 step in IH $CO_2$ gradient.

- In 2016, the IH $CO_2$ gradient is larger than average between Feb-Jun and smaller than average between
Jul-Oct (Fig. 7(a)). These results are again consistent with the behaviour of the dynamical indices. There is reduced IH $CO_2$ eddy transfer in the first half of the year (Fig. 7(b)) but very strong mean transport in the second half of the year (Figs. 7(c) and(d)) that accounts for the annual IH $CO_2$ gradient, as noted in FF18.

- In 1998, the IH $CO_2$ difference exceeds the average from May-Dec and is close to the mean annual cycle
for the rest of the year. We note from Figure 7 that the annual increase in IH $CO_2$ gradient, also shown in Figure 2 of FF18, is largely induced by the Jun-Aug mean Hadley circulation.

- It is suggestive that the relative variation in IH $CO_2$ Feb-May for the three big El Niño years matches that in $u_{duct}$, however it is puzzling that the largest $u_{duct}$ anomaly, 1998, is when IH $CO_2$ is closest to the mean





behaviour. The fact that the mean transport indices at this time of year are also consistently well below

their long term average is also of note, since with $u_{duct}$ close to zero and $-\omega_P$, $-v_P$ indicating descent and

south to north meridional winds, there is no obvious mechanism for IH exchange in this season. Yet, over

the 25 years correlation of the Apr-May IH $CO_2$ peaks with $-\omega_P$, $-v_P$ is significant, $r \approx 0.4$. One possible

explanation for these behaviours in the early part of the Boreal winter/Austral summer may be found in

changes in the volume of the well-mixed portion of the northern hemisphere (see Discussion, Section 5).

Different responses of IH $CO_2$ to wind indices at different ENSO events, and from non-ENSO periods, are

discussed in Section 6.

As an aside, we also include a similar plot for the average SH cgo-spo differences in Figure 8. Despite some

concerns about flask storage at spo, all networks indicate that on average spo baseline $CO_2$ exceeds that at cgo

in the austral summer months. It suggests a possible role for southern ocean uptake influencing cgo, although

the Oct-Feb maximum influence (minimum cgo-spo and sea-air flux) in Nov-Dec appears to precede inversion

estimates of Southern Ocean $CO_2$ uptake south of 30ºS. (Lenton et al., 2013).

**5 Year-to-year variation in the composite records**

The annual net impacts of the various potential influences on site IH $CO_2$ differences (when typical terrestrial

biosphere seasonal variations are balanced) appear in Figure 9.

The largest cgo-spo differences occur in 1992 and 1993. In these years the composite cgo-spo involves

uninterrupted monthly data from two networks only, NOAA and CSIRO. At this time, CSIRO co-measurement

of a subset of NOAA cgo flask samples (Masarie et al., 2001) show CSIRO cgo is around +0.2 ppm higher than

NOAA cgo, too small and in the wrong direction to explain the 1992-1993 network difference, rather favouring

a temporary problem with NOAA spo sampling; no such persisting anomalies of this magnitude have occurred

since. Prior to Mar 1993, NOAA cgo-spo data are lower by around -0.5 ppm compared to CSIRO data

(Supplement Figure S4), with much larger seasonality compared to subsequent NOAA & CSIRO data.

Working through Figure 9 from the left in order to highlight other systematic features:

- Except for 2016, every major El Niño (as indicated by the magnitude of the peak Oceanic Niño Index,
for ONI > 1) corresponds to a transition from low to high IH difference, however the $CO_2$ response is not
proportional to ONI, e.g. comparing 2009-10 to 1997-98, or most noticeably to 2015-16 (the strongest
ONI but the smallest IH $CO_2$ step).

- There is remarkable stability in IH $CO_2$ from 2010-2014; after 2010, there are no significant ONI, and the
5-year increase of ~0.1 ppm is lower than that generally attributed to upward mean fossil fuel emissions
(the 2010-2014 change in FF is 0.73 PgC[-1], which at 0.36 ppm.(PgC)[-1] would result in a 0.26 ppm
increase). There is markedly less variability (composite standard deviation of de-trended annual means
~0.04 ppm) than any equivalent period over the previous 16 years (~0.31 ppm).

- The 2009/2010 year-to-year change of ~0.8 ppm (addressed in FF16 using CSIRO data only) remains the
major year-to-year change in the annual records. Subsequent data now offer an improved perspective on
FF16 conclusions.

- The linear regression through the 25 year mlo-cgo annual data gives a slope of 0.72 ppm yr[-1] compared to
that through monthly values of 0.61 ppm yr[-1] in Figure 3, or through the peaks of 0.09 ppm yr[-1] or the
dips of 0.05 ppm yr[-1]. We interpret this as indicating the combined long-term influence of both eddy and
mean transport on the annual mean IH $CO_2$.



- In 2017, the IH difference is close to the long term mean, with the duct open and Hadley strength
returning to be close to its long-term mean.

**6 Discussion**

The composite monthly IH differences reveal variation from monthly to decadal time scales that exceed
measurement and sampling error (as indicated by the composite standard deviations) thus requiring
biogeochemical explanation. This discussion focusses on the potential of wind indices to explain major features
in IH $CO_2$ variation, with emphasis on periods and events when they are likely to be the dominant influence on
IH $CO_2$. It is intended as a complement to the more general statistical analyses in FF18.

In Figure 6, decreasing $u_{duct}$ acts to lessen eddy IH exchange and increase IH $CO_2$, while the increasing Hadley
circulation (decreasing $v_P$ and $\omega_P$) decreases IH $CO_2$.

The fact that the magnitude of IH $CO_2$ response varies greatly between the 1998, 2010 and 2016 El Niño events
(with little or no eddy transfer occurring in boreal winter/spring in these years) is consistent with a quasi-decadal
variation in the negative excursions of $v_P$ and $\omega_P$ in Figure 6 (most obvious in $\omega_P$). In 1998 and 2010, the
Hadley boreal summer/autumn indices are closer to zero, while 2016 registers an unprecedented negative
excursion.

The complication of IH $CO_2$ variations in the boreal winter/austral summer when $u_{duct}$, $\omega_p$ and $v_P$ indices
indicate that little or no IH exchange occurs (and $u_{duct}$ closure tends to increase mlo $CO_2$), is at a time when the
north to south seasonal variation in the Inter-Tropical-Convergence-Zone is near maximum. If NH peak
terrestrial emissions at that time are diluted into a larger volume of well-mixed NH air, it could offset the mlo
$CO_2$ increase anticipated from $u_{duct}$ closure. This volume effect is likely to be a second order effect in non-El
Niño years.

NH terrestrial biosphere emission anomalies in the 2010-2014 period (Figure 4) are more variable than those in
2000-2005, the opposite of the relative behaviour in IH $CO_2$ variability in Figure 9. These emissions are
relatively small, and frequently occur after the larger IH $CO_2$ anomalies, all inconsistent with a significant
contribution to the composite IH differences; so they are considered second order. The small 2010-2014 trend
(~0.1 ppm compared to 0.26 ppm expected from fossil fuel emissions), and the steadily decreasing westerly
wind strength in $u_{duct}$ over the period, should increase IH $CO_2$ over the fossil fuel trend (FF16). The flattening
trend is consistent with the IH $CO_2$ flux due to IH mixing by the Hadley process overwhelming the increases
expected from Fossil Fuel combustion and from decreasing $u_{duct}$ strength. There is a linear relationship between
$u_{duct}$ and equatorial upper troposphere transient kinetic energy shown in Fig. 6 of Frederiksen and Webster 1988
and discussed in FF16 and FF18). Note that in Figure 6, there is no precedent for similar sustained opposing
behaviour in the two modes of IH transfer. The trend and lack of scatter in 2010-2014 IH $CO_2$ can be understood
by the IH $CO_2$ fluxes being significantly larger than air-surface exchanges at the time.

The magnitude of the IH flux anomalies of up to ~2 PgC month$^{-1}$ exceed known air-surface fluxes in the NH and
are of a magnitude to significantly influence NH $CO_2$ growth rate variability. With increasing fossil fuel fluxes,
the role of IH exchange on the $CO_2$ IH differences, and NH $CO_2$ growth, is expected to become increasingly
important.

The previous inability of carbon cycle models to simulate the 2009/2010 step (FF16, Supplementary
Information) implies that there is inadequate parameterisation of IH $CO_2$ transfer, particularly by eddy



exchange, in some global carbon cycle models. If this is the case, then studies that interpret $CO_2$ behaviour
during ENSO events as a guide to terrestrial biosphere responses to climate will also be compromised. The
ability to simulate the identified features of the composite IH $CO_2$ differences (within the standard deviations)
would provide convincing independent confirmation of atmospheric transport implementation.

**7 Conclusions**

Over the last 25-years there is a high degree of agreement in the measurement of monthly spatial differences in
background $CO_2$ levels by three measurement laboratories using four different sampling methodologies and
sampling frequencies. Geographic isolation of sample collection sites and consistent sophisticated background
selection over the 25 years, as well as coincident monitoring of a wide range of atmospheric species, excludes
local and regional influence on $CO_2$ at mlo, spo and cgo to an extent not generally available at other surface
monitoring sites.
The temporal variation in the composite IH $CO_2$ differences exhibit several systematic features on monthly to
multi-year timeframes that are not reflected in independent evidence of air-surface exchange, but do correspond
to features in dynamical indices selected to represent both eddy and mean IH exchange. The comparisons in this
paper imply a major role for IH exchange of $CO_2$ in NH growth rate variations.
The evidence for a significant influence of atmospheric dynamics on the $CO_2$ IH gradient has relevance for
global carbon cycle studies. It implies that both eddy and mean transport processes, and volume effects, need to
be specifically included in transport model simulations, since the balance between the two is constantly
changing, particularly in El Niño periods when eddy transport is reduced. It also means that El Niño events may
be a poor predictor of the carbon cycle behaviour in non-ENSO years.
Global carbon cycle model simulations should be able to reproduce the major features identified here in the
composite IH records if the re-analyses transport is correctly implemented. In attempting to simulate the
composite differences, one complication is model selection of baseline that matches the flask sampling criteria.
While monthly baseline averages appear to succeed in this respect, a more comprehensive treatment (outside the
scope of this study) based on individual flask measurements rather than monthly averages, and other trace gas
observations FF16, FF18, and in particular radon (Chambers et al., 2016), could possibly improve this process.

*Data availability*. Monthly average NOAA/ESRL, SIO and CSIRO $CO_2$ data were obtained respectively from:
ftp://aftp.cmdl.noaa.gov/data/trace_gases/co2/flask/
http://scrippsco2.ucsd.edu/data/atmospheric_co2/sampling_stations
ftp://pftp.csiro.au/pub/data/gaslab/.These data are also available from the Global Atmosphere Watch database
https://gaw.kishou.go.jp/search.
Ocean Nino Index data were obtained from
https://origin.cpc.ncep.noaa.gov/products/analysis_monitoring/ensostuff/ONI_v5.php
Meteorological data are available from the NOAA/ESRL website at http://www.esrl.noaa.gov/psd/ (Kalnay et
al., 1996)

*The Supplement* related to this article is provided

*Author contributions*.

RJF generated the composite records and their analyses while JSF provided information on atmospheric
dynamics and the roles of transport mechanisms. LPS and RLL contributed $CO_2$ measurement quality
assessments. All four authors contributed to the written document.



***Competing interests.*** The authors declare that they have no conflicts of interest.

***Acknowledgements.***

We thank Ralph Keeling of SIO and Ed Dlugokencky of NOAA for approval of our use of their $CO_2$ data; in
particular we thank Ralph for his suggestion to include SIO data from the $O_2/N_2$ program, and also Brad Hall
from NOAA for information on the early NOAA data. The sustained focus and innovation of CSIRO GASLAB
personnel, plus skilled trace gas sample collection by personnel at the Bureau of Meteorology Cape Grim
Baseline Atmospheric Program, and NOAA personnel at Mauna Loa and South Pole stations underpin this
work. From CSIRO: Vanessa Haverd was generous with her time in providing regional groupings and updates
of the CABLE DVM data; Ying-Ping Wang also provided DVM data; Paul Krummel and Nada Derek advised
on data processing and graphics, and Cathy Trudinger and Rachel Law on the global carbon budget. The
dynamics contributions were prepared using data and software from the NOAA/ESRL Physical Sciences
Division website at http://www.esrl.noaa.gov/psd/



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



**Figure Captions**

**Figure 1**: Monthly $CO_2$ differences (in ppm) from NOAA for sites mlo, cgo and spo and networks SIO1, SIO2 and CSIRO. 5-month running means are highlighted. 1996-2016 mean and standard deviation (bracketed) values are included.

**Figure 2**: Three-month averaged $CO_2$ differences between sites for each of the four sampling networks. (Each 3-month average is plotted on Jan 1 of the appropriate year.

**Figures 3**: Composite station difference data showing the network ensemble average and standard deviation of monthly $CO_2$ for (a) mlo-cgo, (b) mlo-spo and (c) cgo-spo (on a doubly expanded scale). Linear regressions through the IH records are black-dotted lines. Spline polylines link peaks (blue, solid) and dips (red, dotted) of the seasonal IH differences.   Shaded blue rounded rectangles indicate El Niño periods with strongly anomalous equatorial zonal winds.

**Figure 4**: Comparison of the timing and amplitude of terrestrial emissions with variations of the peaks and dips in Figure 3. (a) seasonal anomalies in CABLE emissions and (b) GFED4.1 wild fire seasonal anomalies, for NH (green), EQ (pink) and SH (blue, SH/EQ for GFED4.1).  In (c) The largest anomalies (CABLE NH, CABLE EQ, and GFED4.1 EQ) on the left axis are compared to the ppm variation in peaks (red) and dips (blue) on separate right axes. The axes scaling equates 1 PgC with 1 ppm (see text).

**Figure 5**: Schematic of the boundaries and altitudes of regions used in FF18 to define wind indices that describe eddy IH transfer ($u_{duct}$, westerlies positive) and mean transfer (uplift, negative $\omega_P$) and north to south transfer (negative $v_P$). The shaded area brackets the austral summer extent of the ITCZ in the south (blue dash) and boreal summer extent in the north (red dash).

**Figure 6**: Monthly values of (a) $u_{duct}$, (b) $\omega_P$ and (c) $v_P$. Shading indicates months of enhanced transport which
acts to reduce the IH $CO_2$ difference. Anomalous dynamical periods are highlighted with shaded rectangles.

**Figure 7**: The monthly averages of dynamical factors governing $CO_2$ IH Exchange over the last 25 years. (a) detrended $CO_2$ partial pressure differences mlo-cgo (green), (b) Pacific Eddy transport index $u_{duct}$ (dark blue), (c) Pacific Hadley transport indicated by uplift at 10-15ºN (-$\omega_P$, light red) and (d) North to South transport (-$v_P$, dark red). On average, coincidence of shading in wind indices and shaded months of IH $CO_2$ is a precondition
for increased IH mixing (reduced IH gradient). The more anomalous transport years, 1998 (dots), 2010 (dashes) and 2016 (black line) are shown for each wind index, and for mlo-cgo IH $CO_2$.

**Figure 8**: Composite 25-year average of monthly baseline cgo-spo $CO_2$ (dark blue). Individual network values are shown in orange (NOAA), dark blue (SIO2) and CSIRO (black). Estimates of sea-air $CO_2$ flux seasonality are shown in light blue.

**Figure 9**: Annual changes in the baseline $CO_2$ difference between sites. Interhemispheric differences are plotted on the left axis. The peak magnitudes of the strong El Niños (3-month Nino Region 3.4 average or ONI index) are indicated. The cgo-spo annual differences are plotted on a doubled right-hand scale. Dashed lines represent linear regressions through the annual average data.





**Table**

| # networks | four | three | two | one |
| --- | --- | --- | --- | --- |
| mlo-cgo | - | 277 | 23 | - |
| mlo-spo | 268 | 44 | 2 | - |
| cgo-spo |  | 257 | 42 | 1 |

Table 1: Number of months of data available for composite differences at the baseline sites.



## Figures

**Figure 1**: Monthly $CO_2$ differences (in ppm) from NOAA for sites mlo, cgo and spo and networks SIO1, SIO2 and CSIRO. 5-month running means are highlighted. 1996-2016 mean and standard deviation (bracketed) values are included.

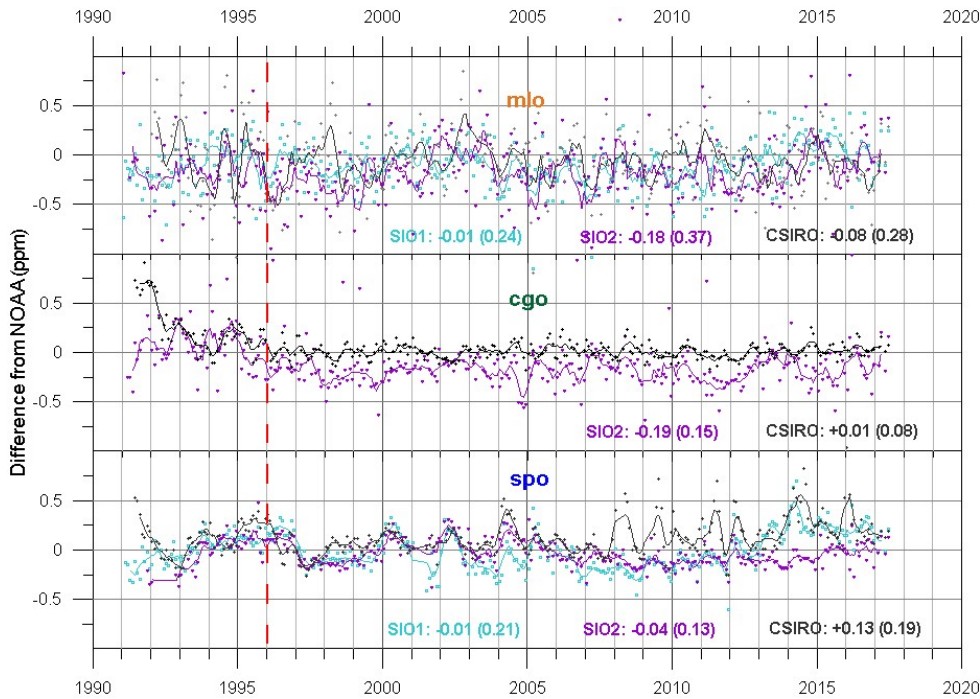



**Figure 2**: Three-month averaged CO$_2$ differences between sites for each of the four sampling networks. (Each
3-month average is plotted on Jan 1 of the appropriate year.

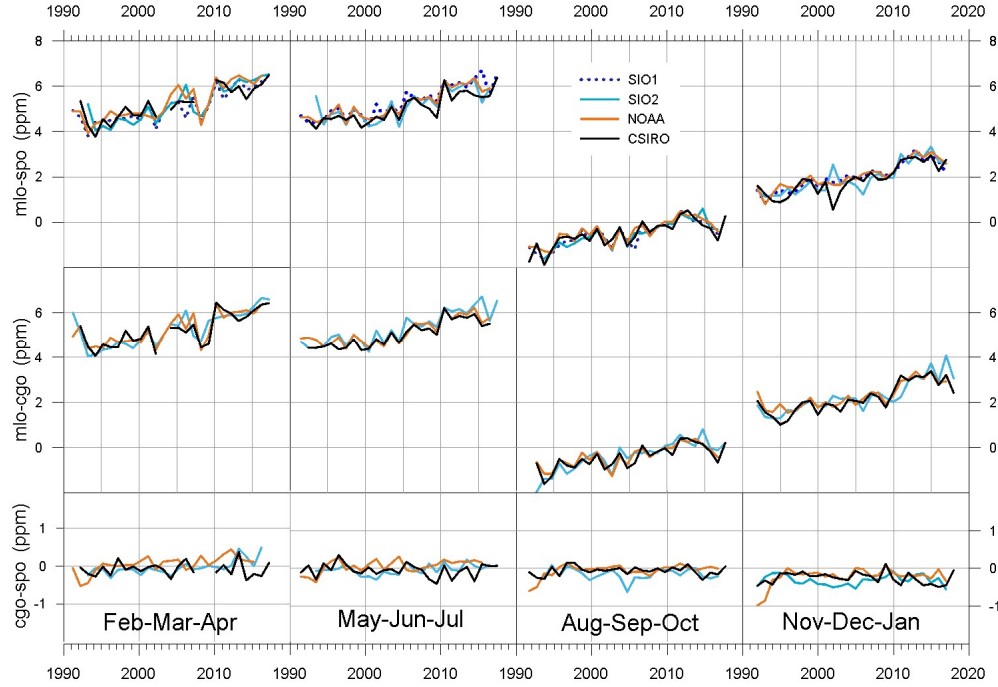





**Figures 3**: Composite station difference data showing the network ensemble average and standard deviation of
monthly $CO_2$ for (a) mlo-cgo, (b) mlo-spo and (c) cgo-spo (on a doubly expanded scale). Linear regressions
through the IH records are black-dotted lines. Spline polylines link peaks (blue, solid) and dips (red, dotted) of
the seasonal IH differences.   Shaded blue rounded rectangles indicate El Niño periods with strongly anomalous
equatorial zonal winds.

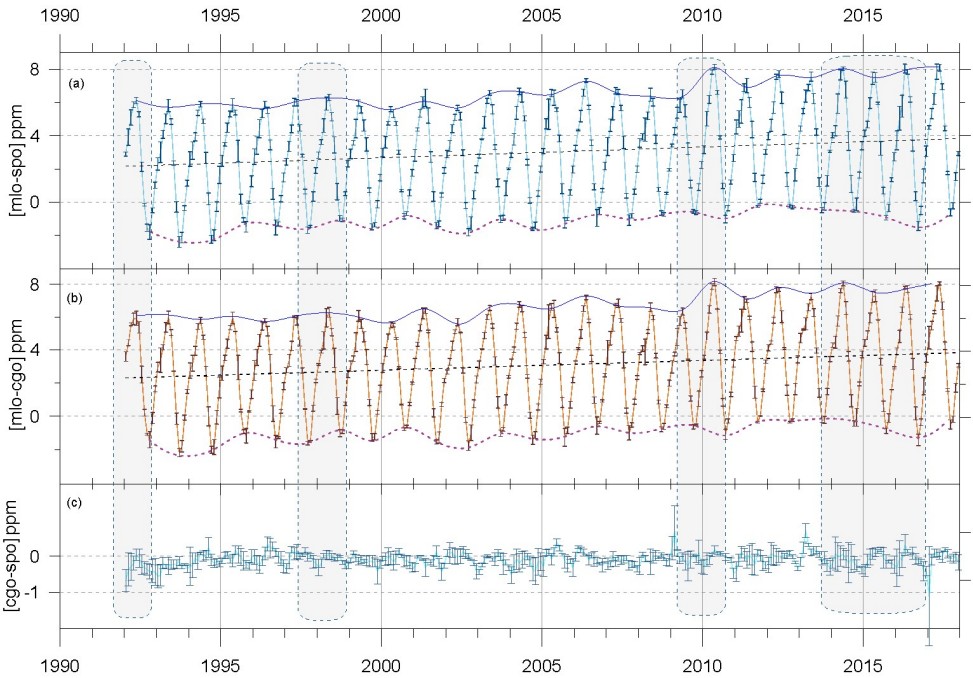




**Figure 4**: Comparison of the timing and amplitude of terrestrial emissions with variations of the peaks and dips in Figure 3. (a) seasonal anomalies in CABLE emissions and (b) GFED4.1 wild fire seasonal anomalies, for NH (green), EQ (pink) and SH (blue, SH/EQ for GFED4.1). In (c) The largest anomalies (CABLE NH, CABLE EQ, and GFED4.1 EQ) on the left axis are compared to the ppm variation in peaks (red) and dips (blue) on 630 separate right axes. The axes scaling equates 1 PgC with 1 ppm (see text).


**Figure 5**: Schematic of the boundaries and altitudes of regions used in FF18 to define wind indices that describe eddy IH transfer ($u_{duct,}$ westerlies positive) and mean transfer (uplift, negative $\omega_P$) and north to south transfer (negative $v_P$). The shaded area brackets the austral summer extent of the ITCZ in the south (blue dash) and boreal summer extent in the north (red dash).


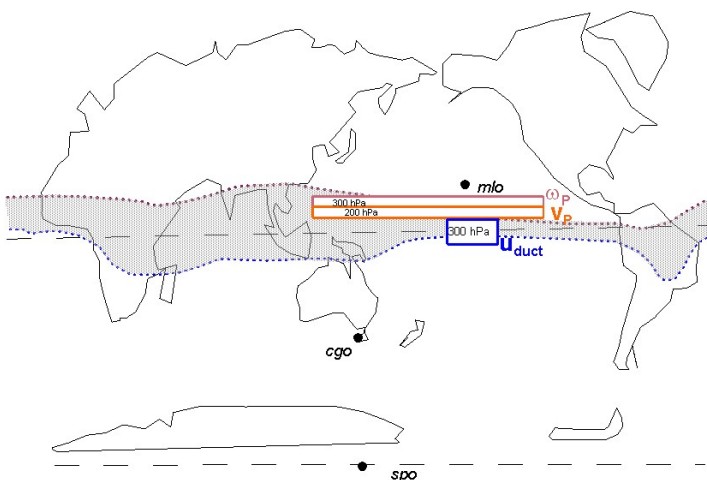



**Figure 6**: Monthly values of (a) $u_{duct}$, (b) $\omega_P$ and (c) $v_P$. Shading indicates months of enhanced transport which acts to reduce the IH $CO_2$ difference. Anomalous dynamical periods are highlighted with shaded rectangles.

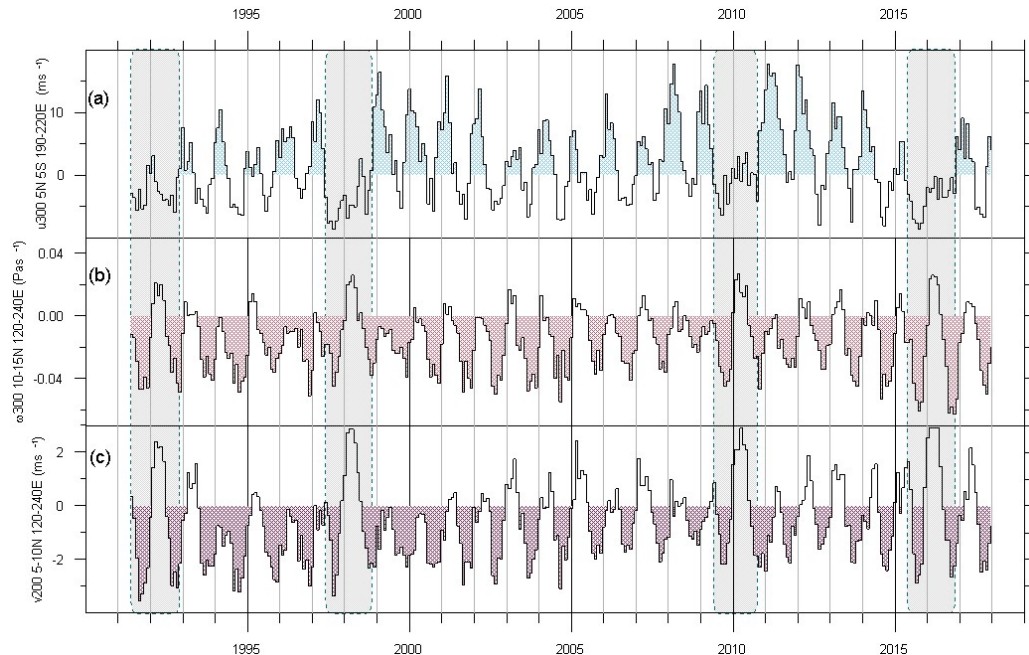





**Figure 7**: The monthly averages of dynamical factors governing $CO_2$ IH Exchange over the last 25 years. (a) detrended $CO_2$ partial pressure differences mlo-cgo (green), (b) Pacific Eddy transport index $u_{duct}$ (dark blue), (c) Pacific Hadley transport indicated by uplift at 10-15ºN (-$\omega_P$, light red) and (d) North to South transport (-$v_P$, dark red). On average, coincidence of shading in wind indices and shaded months of IH $CO_2$ is a precondition for increased IH mixing (reduced IH gradient). The more anomalous transport years, 1998 (dots), 2010 (dashes) and 2016 (black line) are shown for each wind index, and for mlo-cgo IH $CO_2$.

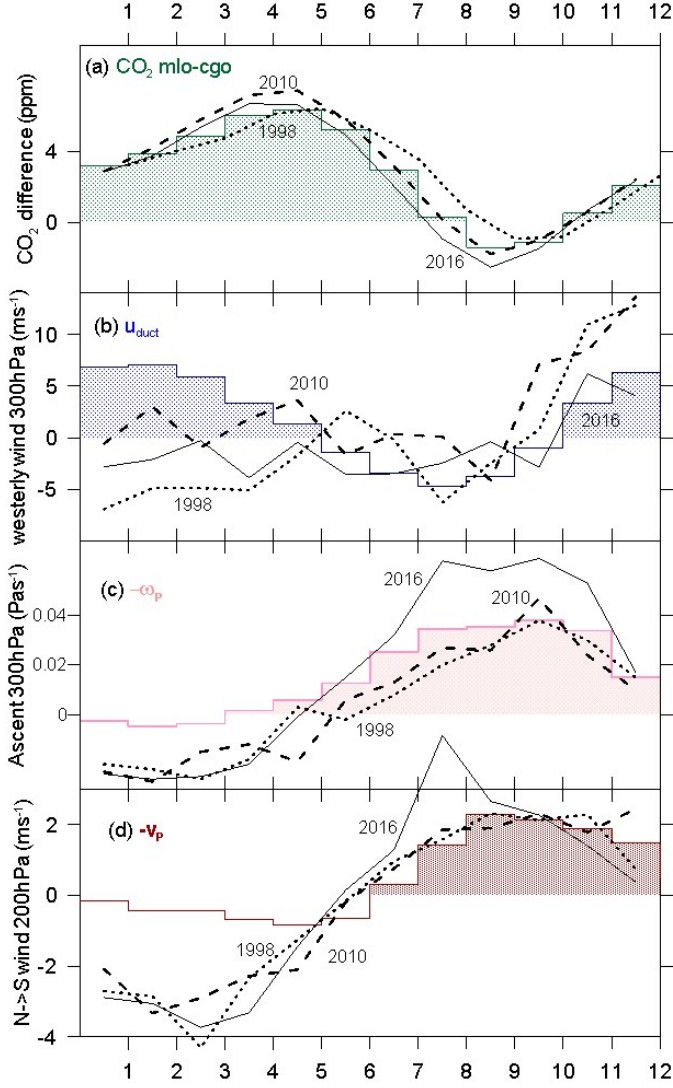



**Figure 8**: Individual network values that contribute to the composite are shown in orange (NOAA), dark blue
(SIO2) and CSIRO (black). Estimates of sea-air $CO_2$ flux seasonality are shown in light blue.

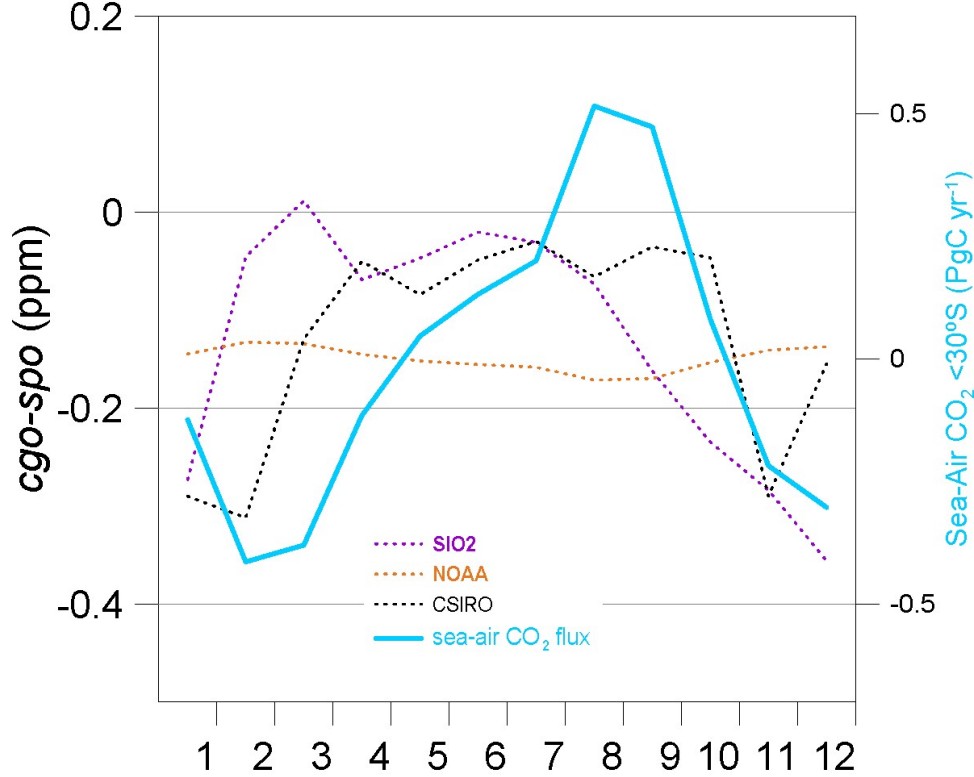




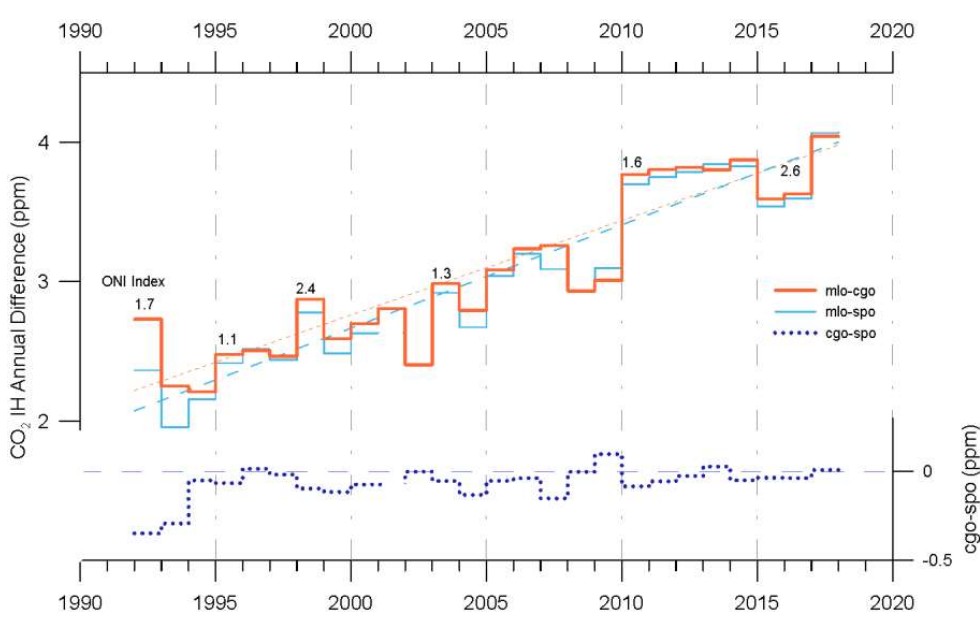

**Figure 9**: Annual changes in the baseline $CO_2$ difference between sites. Interhemispheric differences are plotted on the left axis. The peak magnitudes of the strong El Niños (3-month Nino Region 3.4 average or ONI index) are indicated. The cgo-spo annual differences are plotted on a doubled right-hand scale. Dashed lines represent
linear regressions through the annual average data.