# Peer review of "Variability in a four-network composite of atmospheric CO2 differences between three primary baseline sites"

_Atmospheric Chemistry and Physics, 2019_

## Referee Comment (RC1) · Anonymous Referee #1 · 3 Jul 2019

The article by Francey et al describes and analyses 25-year composites of interhemispheric (IH) baseline CO2 differences from NOAA, CSIRO and two independent SIO analysis. They show a good agreement between the 4 monitoring networks and explore the influence of atmospheric dynamics on the CO2 IH gradient with a focus on El Niño periods. The results highlight the importance of IH CO2 transfer parametrization in global carbon cycle models.

In general, the paper is scientifically sound and worthy of publication; however, the writing needs some modification and improvement. There is also several typos regarding the results hence the paper needs a careful reading/checking. After addressing the

comments, the manuscript will be suitable for publication.

General Comments:

1. Currently the introduction sounds like this paper is submitted to AMT and not ACP, and it does not entirely sound like an introduction. For moments it is too techical and it is hard to understand the aim of the work. I recommend re-framing the introduction, to better attract the readers that are interested about this work. I would suggest to include an overview of the key findings of FF16 and FF18 (relevant to the work in this paper), and higlight better what this work aims to add to the previous findings. Parts of referring to results in FF16 and FF18 are indeed included later in the paper; however, the way/order they are represented is making things hard to link together and to see the big picture (and it makes the paper harder to read). Potentially the advantages/disadvantages in the Introduction could be added into a separate section (or combine with Section 2).

2. Usually Introductions end with a paragraph summarizing the aim of the work. I had the feeling that this comes quite late at the end of Section 2. In addition to re-framing the Introduction I would move the last paragraph from Section 2 to the Introduction. This would also partially solve the missing key findings of FF16 and FF18.

3. At the moment the clarification/discussion of few things that can potentially lead to confusion for the readers are missing from the paper. It would be good to just summarize somewhere in the paper: 1) the important ENSO periods discussed in the analysis. The reason for this is that I kept thinking that the highlighted 2010 period in Figure 3 had potentially something to do with the strong 2011 La Niña event (although without reading FF16 first). 2) Why La Niña is not discussed at all and how it would impact IH (e.g., La Niña periods facilitates interhemispheric mixing of trace gases while El Niño inhibits interhemispheric exchange... ). In Section 4 the authors wrote "Different responses of IH $CO_2$ to wind indices at different ENSO events, and from non-ENSO periods, are discussed in Section 6." and since ENSO includes both El Niño and La

[Figure]

Niña, the impact of La Niña should be at least mentioned somewhere. These things are discussed in FF16 and FF18; however, it would be beneficial to add a sentence or two here also to be easier to track/understand the results.

4. Page 1 line 24 and elsewhere in the paper: "5-year relatively ENSO-quiet period" - I assume the authors meant "5-year relatively El Niño-quiet period" since 2011 was a strong La Niña period, so ENSO-quiet period is misleading.

5. Figure 4 - are there emission anomaly uncertainties that could be added to the Figure and included in the discussion in Section 4.1?

Specific Comments:

* Section 2 is quite lengthy and could be simplified. A careful reading to condense some of the text would be useful.

* Page 5 line 183: "e.g. 256 of 300 months have 4 networks" - It seems like the Table shows 268 instead of 256 for mlo-spo, or I misunderstood the Table in which case the authors need to give a better explanation of the Table. Moreover, for mlo-spo with both 268 or 256 the total number of months does not add up to 300. Also regarding the Table, what is the difference between empty boxes and the ones with '-'. If nothing then use consistent marking.

* Page 7 line 228: "It is assumed here that flux variations from ocean sources are much smaller than terrestrial" - it would be good to have some references here. Also just mention why anthropogenic (fossil fuel) emissions are not compared here.

* Page 9 line 334: It would be good to state why did the authors choose the ONI index as the ENSO index instead of the other indices. It would be also interesting to see if other ENSO indices show the same results (but not strictly necessary to include in this paper if it is too time consuming).

* Page 9 line 337: "there are no significant ONI" - well relative to El Niño but not relative to La Niña, so the statement is a little bit misleading.

Technical Comments:

* Page 2 1st and 2nd paragraph: It feels like a weird jump between the two paragraphs, a 'linking' sentence between the two would be good.

* Page 5 line 166, Page 8 line 308, Page 9 line 334: however -> ;however,

* Page 6 line 200: & -> and

* Page 8 line 271: FF18.These -> missing space before These

* Figure 5: ITCZ -> Inter Tropical Convergence Zone, the abbreviation was not defined

* Page 9 line 314: "see Discussion, Section 5" -> see Discussion, Section 6 maybe?

* Page 9 line 339 ppm.(PgC) -> missing space before (PgC)

* Page 10 line 258-359: why is this a separate paragraph?

* northern hemisphere is somewhere capitalized and somewhere not in the text

* NH abbreviation is not defined

* Figure 3 inconsistency between figure and caption: (a) mlo-cgo, (b) mlo-spo -> (a) mlo-spo, (b) mlo-cgo

* Figure 7 and 8 please add x axis label

---

## Referee Comment (RC2) · Anonymous Referee #2 · 16 Jul 2019

Summary:

Francey et al. present an analysis of monthly data for three measurement stations (MLO, CGO and SPO) to investigate the interhemispheric CO2 difference (IH CO2, defined as concentration difference of MLO-CGO or MLO-SPO) variations over the last 25 years. After comparing the different data sets, the potential biogeochemical drivers for the short-term, seasonal and inter-annual variability of IH CO2 is discussed. The manuscript highlights the ability of different atmospheric transport indices to explain these IH CO2 variations. The manuscript is well-structured and contains important and refined ideas that will be useful in guiding the improvement of GCMs commonly used

in global carbon cycle modelling. Its topic is well suited for ACP and surely of interest to the wider scientific community. However, two general comments and a few minor comments should be addressed before publication.

General comments:

One key argument of this paper is that the variations of IH CO2 cannot be explained by the net uptake of CO2 in the NH terrestrial biosphere. Unfortunately, this assumption is based only on a single DGVM, which are known to be highly uncertain. A comparison of CABLE and other models can be found here: https://journals.ametsoc.org/doi/pdf/10.1175/2008JCLI2378.1. Multiple DGVMs or an ensemble mean of CMIP LSMs would be more robust here. When considering fluxes from optimized emission products (e.g. CarbonTracker-EU) a significantly larger share of IH CO2 could be explained by the NH terrestrial biosphere, although likely not contradicting the finding that IH CO2 is strongly influenced by transport processes. The manuscript does not use consistent language around the main driver of NH CO2 enhancements. The claim that respiration is maximal in early parts (FEB-APR MAY-JUL) seems unlikely and needs to be supported. However, later the authors refer to NBP and "terrestrial emissions" (which could likely peak in different seasons than autotrophic & heterotrophic respiration. As this is a key issue a clarification would be most helpful.

Specific comments:

Line 31 and Line 44: Why is the proposed method only compared to growth rate studies here and not to 4D-VAR and ensemble kalman filter/smoother data assimilation systems that have been shown to reproduce global scale fluxes with fairly reasonable performance?

Line 55: Figure 1 seems to suggest that with-in network errors do not cancel our. No consistent offset between the different measurement programs was found for all sites. However, to assume that the difference is small compared to the IH CO2 signal seems perfectly reasonable.

Line 60: Multi-species observations are mentioned here, but not discussed in this manuscript. Examples of relevant species and isotopes should be given (or paragraph removed).

Line 63+: The authors do not mention the added value of quasi-continuous data from in-situ observations. If they were available (or used here), issues relating to the impact of different sampling dates and sampling frequencies of the different flask programs could assessed more quantitatively .

Line 122: Unclear what kind of information is available or referenced in Keeling 1998. It would be great to have more details .

Line 131. Please clarify: the maximum of 7-10ppm of IH CO2 occurs when most exchange occurs between NH and SH? Seems counter-intuitive or do you talk about the specific gradients of two sites here?

Line 133: Do the offsets really "cancel" here? (see Line 55).

Line 155: Why are those definitions of seasons used here and not the more common meteorological or astronomical definitions? Please consider clarifying.

Line 167: Why was the decision made not corrected or flag the flask data but "average out" potential outliers by making a composite time series?

Line 172: Please provide a reference on the assumption that maximum respiration from NH forests occurs in the seasons in FEB-APR and MAY-JUL? This seems fairly unlikely, as especially FEB-APR is still cold in most of the boreal forest regions in the NH and both autotrophic and heterotrophic respiration typically increase later in the year e.g. in (late) summer. Or please clarify if this refers to net biome productivity or net ecosystem exchange or "terrestrial emissions" (mentioned later in the manuscript).

Line 183: The data in table 1 seems inconsistent with the example here. In general, 1992-2016 is mentioned, but later 1992-2017 is shown. (Figure in supplement has 288 months of maximum data, main text figure refers to 300 months maximum).

[Figure]

Line 190: Please clarify "generally attributed to NH forest..." this manuscript argues (in later sections) that 1.) the IH $CO_2$ gradient is dominated by transport and that 2.) the variability in the IH $CO_2$ is driven by the MLO time series. Is this consistent?

Line 199: Why is 2017 now included (see comment L183)

Line 214: Some technical detail on how the fit was done would be most useful to the reader here, did you perform a sensitivity analysis for different types of splines?

Line 220: Please clarify: the assumption that the global atmospheric $CO_2$ budget is not driven by IH transport but by exchanges to other compartments (biosphere, oceans, etc.) does not have to rely on assuming that transport is correct. The total global atmospheric $CO_2$ budget is the same no matter if $CO_2$ is in the NH or SH. I assume the authors want to argue that the NH versus SH budget might be significantly wrong when not accounting for IH exchange or maybe that the sites used (e.g. MLO) reflect more signals than just emissions and sinks in NH?

Line 235+: Why was the CABLE model used. DGVMs are inherently uncertain, so using multiple DGVMs or an ensemble estimate would give a better representation of the range of estimates of NH terrestrial fluxes. Optimized emission products, e.g. CarbonTracker-Europe report significantly higher NH terrestrial uptake (same order of magnitude as IH processes discussed later in the manuscript and in Figure 4). Available for download at: http://www.carbontracker.eu/fluxtimeseries.php [See also general comment]

Line 253: The qualitative study on the potential impact of $CO_2$ emissions on global and hemispheric $CO_2$ concentrations seems very instructive. However, a more qualitative estimate using a GCM could be beneficial here.

Line 286: The authors raise an interesting point here: the MLO-SPO and MLO-CGO data is "effectively identical". Why was SPO data included in this study? A separate study of CGO-SPO could have been more enlightening if it could address the question

of CO2 uptake in the Southern Oceans.

Line 345: What are the uncertainties of the estimated annual concentration trends and are they statistically significant (and different)?

Line 368: Here, the manuscript refers to "terrestrial emissions" is this equal to F(ffco2)+NBP?

Line 404: How would the IHCO2 impact the growth rate of CO2 derived at other long-term reference sites in the NH. Like Barrow, US and Alert, CA? Could analyzing data from those sites help to further separate the impact of IH mixing versus NH fluxes?

———————————————————

---

## Author Comment (AC1) · 26 Aug 2019

Anonymous Referee #1 THE RESPONSES REFER TO LINE NUMBERS IN A REVISED DOCUMENT ATTACHED HERE AS A SUPPLEMENT.

The article by Francey et al describes and analyses 25-year composites of nterhemispheric (IH) baseline CO2 differences from NOAA, CSIRO and two independent SIO analysis. They show a good agreement between the 4 monitoring networks and explore the influence of atmospheric dynamics on the CO2 IH gradient with a focus on El Niño periods. The results highlight the importance of IH CO2 transfer parametriza-

tion in global carbon cycle models. In general, the paper is scientifically sound and worthy of publication; however, the writing needs some modification and improvement. There is also several typos regarding the results hence the paper needs a careful eading/checking. We have modified, hopefully improved, and more carefully checked, the writing. After addressing the comments, the manuscript will be suitable for publication. General Comments:

General Comment 1. Currently the introduction sounds like this paper is submitted to AMT and not ACP, and it does not entirely sound like an introduction. For moments it is too techical and it is hard to understand the aim of the work. I recommend re-framing the introduction, to better attract the readers that are interested about this work. I would suggest to include an overview of the key findings of FF16 and FF18 (relevant to the work in this paper), and higlight better what this work aims to add to the previous findings. Parts of referring to results in FF16 and FF18 are indeed included later in the paper; however, the way/order they are represented is making things hard to link together and to see the big picture (and it makes the paper harder to read). Potentially the advantages/disadvantages in the Introduction could be added into a separate section (or combine with Section 2).

RESPONSE. A substantial rewrite of the Introduction is in response to these recommendations. Technical discussion is moved to Section 2 or addressed in an expanded Supplement. The Scientific 'big picture' should be clearer, and the different roles of FF16, FF18 and this paper are more clearly specified. The advantages and disadvantages are important when aiming for internal consistency over three decades. However, because of REF1 comments this discussion is now in the Supplement.

General Comment 2. Usually Introductions end with a paragraph summarizing the aim of the work. I had the feeling that this comes quite late at the end of Section 2. In addition to re-framing the Introduction I would move the last paragraph from Section 2 to the Introduction. This would also partially solve the missing key findings of FF16 and FF18.

RESPONSE: The final paragraph of the introduction (Line 68) now defines the scope of the paper.

General Comment 3. At the moment the clarification/discussion of few things that can potentially lead to confusion for the readers are missing from the paper. It would be good to just summarize somewhere in the paper: 1) the important ENSO periods discussed in the analysis. The reason for this is that I kept thinking that the highlighted 2010 period in Figure 3 had potentially something to do with the strong 2011 La Niña event (although without reading FF16 first). 2) Why La Niña is not discussed at all and how it would impactIH(e.g., LaNiña periods facilitates interhemispheric mixing of trace gases while ElNiño inhibits interhemispheric exchange... ). In Section4 the authors wrote "Different responses of IH CO2 to wind indices at different ENSO events, and from non-ENSO periods, are discussed in Section 6." and since ENSO includes both El Niño and La Niña, the impact of La Niña should be at least mentioned somewhere. These things are discussed in FF16 and FF18; however, it would be beneficial to add a sentence or two here also to be easier to track/understand the results.

RESPONSE: The concerns about La Nina are mentioned in the Abstract " (line 25)", in lines 63-66 of the Introduction and elsewhere the term ENSO is used to include both El Niño and La Niña events. The ONI in strong La Nina years are now included in Figure 9.

General Comment 4. Page 1 line 24 and elsewhere in the paper: "5-year relatively ENSO-quiet period" I assume the authors meant "5-year relatively El Niño-quiet period" since 2011 was a strong La Niña period, so ENSO-quiet period is misleading.

RESPONSE: Agreed, see point 3 for the revised approach

General Comment 5. Figure 4 - are there emission anomaly uncertainties that could be added to the Figure and included in the discussion in Section 4.1?

RESPONSE: To address this point, we have added a 16 DVM model composite, and seasonal Fossil data to Figure 4 to demonstrate their relative magnitudes. Because all surface-to-air fluxes appear very much smaller than required to significantly alter the baseline CO2 differences, we have not dwelt on obtaining uncertainties from source data.

Specific Comments:

* Section 2 is quite lengthy and could be simplified. A careful reading to condense some of the text would be useful. RESPONSE: A new section 3 is introduced, and the sampling advantages/disadvantages moved to Supplement S3 in order to break up the lengthy section.

* Page 5 line 183: "e.g. 256 of 300 months have 4 networks" - It seems like the Table shows 268 instead of 256 for mlo-spo, or I misunderstood the Table in which case the authors need to give a better explanation of the Table. Moreover, for mlospo with both 268 or 256 the total number of months does not add up to 300. Also regarding the Table, what is the difference between empty boxes and the ones with'-'. If nothing then use consistent marking.

RESPONSE The Table is now updated to 2017 (since SIO spo data became available). The numbers are now consistent, and the main text adjusted accordingly.

* Page 7 line 228: "It is assumed here that flux variations from ocean sources are much smaller than terrestrial" - it would be good to have some references here. Also just mention why anthropogenic (fossil fuel) emissions are not compared here.

RESPONSE: This is now addressed in lines 216, lines 315-320, and Figure 8. The seasonal variation in Southern Ocean fluxes (an important part of the global ocean flux) are similar to terrestrial fluxes from the region but much smaller than NH terrestrial flux variation. High precision continuous CO2 monitoring across the Southern Ocean (Stavert et al., 2019, and personal communication) confirm relatively small variation. Fossil emission seasonal variations are included line 322 and in the new Figure 4(d).

* Page 9 line 334: It would be good to state why did the authors choose the ONI index as the ENSO index instead of the other indices. It would be also interesting to see if other ENSO indices show the same results (but not strictly necessary to include in this paper if it is too time consuming).

RESPONSE: We have re-examined the ONI data and they give essentially the same result as Nino3 and Nin3.4 indices, as stated in line 331.

*Page9 line337: "there are no significant ONI"-well relative to ElNiño but not relative to La Niña, so the statement is a little bit misleading.

RESPONSE: See reworded lines 3333-336.

Technical Comments:

*Page2 1st and 2nd paragraph: It feels like a weird jump between the two paragraphs, a 'linking' sentence between the two would be good.

RESPONSE: A linking sentence is provided, lines 33-36.

* Page 5 line 166, Page 8 line 308, Page 9 line 334: however -> ;however, * Page 6 line 200: & -> and * Page 8 line 271: FF18.These -> missing space before These * Figure 5: ITCZ -> Inter Tropical Convergence Zone, the abbreviation was not defined * Page 9 line 314: "see Discussion, Section 5" -> see Discussion, Section 6 maybe? * Page 9 line 339 ppm.(PgC) -> missing space before (PgC) * Page 10 line 258-359: why is this a separate paragraph? * northern hemisphere is somewhere capitalized and somewhere not in the text * NH abbreviation is not defined * Figure 3 inconsistency between figure and caption: (a) mlo-cgo, (b) mlo-spo -> (a) mlo-spo, (b) mlo-cgo * Figure 7 and 8 please add x axis label

RESPONSE: Thank you, these 10 items have been corrected

Please also note the supplement to this comment:
https://www.atmos-chem-phys-discuss.net/acp-2019-300/acp-2019-300-AC1- supplement.pdf

[revised manuscript text omitted]

---

## Author Comment (AC2) · 26 Aug 2019

NOTE: THE LINE NUMBERS IN RESPONSES REFER TO A REVISED MAIN TEXT ATTACHED AS A SUPPLEMENT TO THESE COMMENTS.

Anonymous Referee #2 Summary: Francey et al. present an analysis of monthly data for three measurement stations (MLO, CGO and SPO) to investigate the interhemispheric CO2 difference (IH CO2, defined as concentration difference of MLO-CGO or MLO-SPO) variations over the last 25 years. After comparing the different data sets, the potential biogeochemical drivers for the short-term, seasonal and inter-annual variability of IH CO2 is discussed. The manuscript

highlights the ability of different atmospheric transport indices to explain these IH CO2 variations. The manuscript is well-structured and contains important and refined ideas that will be useful in guiding the improvement of GCMs commonly used in global carbon cycle modelling. Its topic is well suited for ACP and surely of interest to the wider scientific community. However, two general comments and a few minor comments should be addressed before publication.

General comments: One key argument of this paper is that the variations of IH CO2 cannot be explained by the net uptake of CO2 in the NH terrestrial biosphere. Unfortunately, this assumption is based only on a single DGVM, which are known to be highly uncertain. A comparison of CABLE and other models can be found here: https://journals.ametsoc.org/doi/pdf/10.1175/2008JCLI2378.1. Multiple DGVMs or an ensemble mean of CMIP LSMs would be more robust here.

RESPONSE: A 16-DGVM ensemble of extra-tropical NBP from Bastos et al. 2018 is now included (lines222-225), including representation in Figure 4. It confirms the surface flux anomalies are too small to account for IHDCO2 changes.

When considering fluxes from optimized emission products (e.g. CarbonTracker-EU) a significantly larger share of IH CO2 could be explained by the NH terrestrial biosphere, although likely not contradicting the finding that IH CO2 is strongly influenced by transport processes.

RESPONSE: We do not discuss air-surface fluxes derived from CO2 data that are less spatially representative, and/or rely on atmospheric transport modelling. The latter introduce additional model degrees of freedom and potentially overestimate terrestrial variability if the variability in atmospheric IH transport is not adequately captured" (lines 224-227).

The manuscript does not use consistent language around the main driver of NH CO2 enhancements. The claim that respiration is maximal in early parts (FEB-APR MAY-JUL) seems unlikely and needs to be supported. However, later the authors refer

to NBP and "terrestrial emissions" (which could likely peak in different seasons than autotrophic & heterotrophic respiration. As this is a key issue a clarification would be most helpful.

RESPONSE: This is a valid criticism. The scope of this paper is to explore and seek explanation for variations in the baseline $CO_2$ records. While there are clear implications for the global carbon budget, lack of advanced knowledge in every individual component of the budget limits a more comprehensive approach. Reducing uncertainty in the critical atmospheric component of the global budget is the purpose of this study. This is clarified in Lines 68-73 of the rewritten Introduction an in responses to Specific comment Lines 31, 131, 172, 190.

Specific comments: Line 31 and Line 44: Why is the proposed method only compared to growth rate studies here and not to 4D-VAR and ensemble kalman filter/smoother data assimilation systems that have been shown to reproduce global scale fluxes with fairly reasonable performance?

RESPONSE: The Introduction rewrite aims to clarify this. See also lines 222-225 generated in response to General comments.

Line 55: Figure 1 seems to suggest that with-in network errors do not cancel our. No consistent offset between the different measurement programs was found for all sites. However, to assume that the difference is small compared to the IH $CO_2$ signal seems perfectly reasonable.

RESPONSE: We make this assumption.

Line 60: Multi-species observations are mentioned here, but not discussed in this manuscript. Examples of relevant species and isotopes should be given (or paragraph removed).

RESPONSE: Multi-species studies, outside the current scope, have the potential to further inform this debate. They are briefly mentioned at line 63 and in the expanded

Supplementary information.

Line 63+: The authors do not mention the added value of quasi-continuous data from in-situ observations. If they were available (or used here), issues relating to the impact of different sampling dates and sampling frequencies of the different flask programs could assessed more quantitatively.

RESPONSE: The advantages and disadvantages are relevant when aiming for internal consistency over three decades. Challenges to achieving consistent site differences are much greater with the conventional NDIR analysers used over most of this period. This discussion is now focussed in the new Supplement. In relation to spatial differences, NDIR in situ analyses have had calibration limitations due to short lifetime of reference and calibration gases. Flasks analyses in a central laboratory have largely avoided these limitations over decadal timeframes.

Line 122: Unclear what kind of information is available or referenced in Keeling 1998. It would be great to have more details.

RESPONSE: It is a brief reflection of the difficulties in maintaining government support for monitoring. It is best articulated in the original publication.

Line 131. Please clarify: the maximum of 7-10ppm of IH $CO_2$ occurs when most exchange occurs between NH and SH? Seems counter-intuitive or do you talk about the specific gradients of two sites here?

RESPONSE: Yes, poorly worded. A $CO_2$ partial pressure difference between hemispheres is a prerequisite for net IH exchange. See lines 120-1.

Line 133: Do the offsets really "cancel" here? (see Line 55).

RESPONSE: The words used were "largely cancel", which is the case. (It is supported in Figure 4 by the ïĄĎIHCO2 response to the 1997 GFED anomaly).

Line 155: Why are those definitions of seasons used here and not the more common

meteorological or astronomical definitions? Please consider clarifying.

RESPONSE: The reasons are stated in lines 141-143: "the particular 3-month seasonal selection distinguishes periods of relatively stable partial pressure differences between hemispheres and the selected seasons also distinguish eddy and mean IH transport mechanisms (FF18)"

Line 167: Why was the decision made not corrected or flag the flask data but "average out" potential outliers by making a composite time series?

RESPONSE: This sentence generally describes selection made by each laboratory prior to publication of the monthly averages used here. In the subsequent compositing this data clear statistical outliers are revealed of unknown origin. In the composite these are suppressed by averaging. See lines 115-118.

Line 172: Please provide a reference on the assumption that maximum respiration from NH forests occurs in the seasons in FEB-APR and MAY-JUL? This seems fairly unlikely, as especially FEB-APR is still cold in most of the boreal forest regions in the NH and both autotrophic and heterotrophic respiration typically increase later in the year e.g. in (late) summer. Or please clarify if this refers to net biome productivity or net ecosystem exchange or "terrestrial emissions" (mentioned later in the manuscript).

RESPONSE: See response at the end of General Comments above. The wording at line 157 now merely states the widely accepted general reason for the NH $CO_2$ seasonality.

Line 183: The data in table 1 seems inconsistent with the example here. In general, 1992-2016 is mentioned, but later 1992-2017 is shown. (Figure in supplement has 288 months of maximum data, main text figure refers to 300 months maximum).

RESPONSE: The Table is now updated to 2017 (since SIO spo data became available). The numbers are now consistent, and the main text adjusted accordingly.

Line 190: Please clarify "generally attributed to NH forest..." this manuscript argues (in

later sections) that 1.) the IH CO2 gradient is dominated by transport and that 2.) the variability in the IH CO2 is driven by the MLO time series. Is this consistent?

RESPONSE: The seasonality in mlo CO2 is generally attributed to NH forests, and SH seasonality is small by comparison. However, the anomalies (mean seasonality subtracted) correspond primarily to anomalies in IH transport indices.

Line 199: Why is 2017 now included (see comment L183)

RESPONSE: See reply to comment L183.

Line 214: Some technical detail on how the fit was done would be most useful to the reader here, did you perform a sensitivity analysis for different types of splines?

RESPONSE: Spline polylines, generally available in commercial plotting software, link peaks and dips and serve only to visually aid discussion.

Line220: Please clarify: the assumption that the global atmospheric CO2 budget is not driven by IH transport but by exchanges to other compartments (biosphere, oceans, etc.) does not have to rely on assuming that transport is correct. The total global atmospheric CO2 budget is the same no matter if CO2 is in the NH or SH. I assume the authors want to argue that the NH versus SH budget might be significantly wrong when not accounting for IH exchange or maybe that the sites used (e.g. MLO) reflect more signals than just emissions and sinks in NH?

RESPONSE: Yes. Wording in Section 5 is adjusted to make this clearer.

Line 235+: Why was the CABLE model used. DGVMs are inherently uncertain, so using multiple DGVMs or an ensemble estimate would give a better representation of the range of estimates of NH terrestrial fluxes. Optimized emission products, e.g. CarbonTracker-Europe report significantly higher NH terrestrial uptake (same order of magnitude as IH processes discussed later in the manuscript and in Figure 4). Available for download at: http://www.carbontracker.eu/fluxtimeseries.php [See also general comment]

RESPONSE: These issues are addressed in the response to General comments, above.

Line253: The qualitative study on the potential impact of CO2 emissions on global and hemispheric CO2 concentrations seems very instructive. However, a more qualitative estimate using a GCM could be beneficial here.

RESPONSE: Maybe, if the GCM has sufficient resolution and upper equatorial tropical parameterisation? It is outside the scope of this study.

Line 286: The authors raise an interesting point here: the MLO-SPO and MLO-CGO data is "effectively identical". Why was SPO data included in this study? A separate study of CGO-SPO could have been more enlightening if it could address the question of CO2 uptake in the Southern Oceans.

RESPONSE: This point is specifically addressed in the new introduction. CO2 data quality (measurement and spatial representation) is considered a key issue in this paper, in which the composite plays a central role. The cgo-spo comparison is the best independent evidence for both factors. Note: The Southern Ocean uptake is informed by other records (including in ultra-high precision in situ monitoring at cgo and Macquarie Island plus other Antarctic sites). This is outside the scope of this paper and is the subject of on-going investigation by others (e.g. Stavert et al.). And we speculate that other factors such as seasonality in FF emissions may be involved, see line 315-322.

Line 345: What are the uncertainties of the estimated annual concentration trends and are they statistically significant (and different)?

RESPONSE: Statistical uncertainties in trends, are now included throughout the paper. We also provide an estimate of uncertainty in the annual average data of Figure 9, Line 326.

Line 368: Here, the manuscript refers to "terrestrial emissions" is this equal to

F(ffco2)+NBP?

RESPONSE: Yes, text amended, Line 367.

Line 404: How would the IHCO2 impact the growth rate of CO2 derived at other long term reference sites in the NH. Like Barrow, US and Alert, CA? Could analyzing data from those sites help to further separate the impact of IH mixing versus NH fluxes?

RESPONSE: Possibly, but spatial representation and quality of data are limiting factors.

Please also note the supplement to this comment:
https://www.atmos-chem-phys-discuss.net/acp-2019-300/acp-2019-300-AC2-supplement.pdf

[revised manuscript text omitted]

---

## Author Response (AR1)

*Interactive comment on Atmos. Chem. Phys. Discuss., https://doi.org/10.5194/acp-2019-300, 2019.*

**Comments for Editor**

The early sections of the Discussion paper have been substantially restructured to follow the suggestions of REF1. The WORD track changes feature shows the changes.

The technical detail in the introductory sections has been reduced:

- Flask Sampling Advantages and Disadvantages is moved to an expanded Supplement (S3).
- The original Section 2 is divided into two sections
  - 2 Background Information on Flask Networks, and
  - 3 Network Inter-comparison
- The scope of the study is more clearly defined at the end of the Introduction – it puts the focus more clearly on variation in the baseline $CO_2$ records, and their likely relevance to the global carbon budget rather than attempting to revise the carbon budget.

Two new references, and text have been added to address concerns of REF2:

- Adding Bastos et al. (2018) addresses concerns about using only one DVM. Bastos s et al. results are now included in Figure 4 (and prove equally insignificant compared to atmospheric interhemispheric fluxes). Figure 4(d) is added, which now compares the seasonal variability of the contributing processes to CO2 IH differences, and reinforces the basic point of our paper.
- Adding Stavert et al. (2019) contributes to addressing concern about using both mlo-cgo and mlo-spo, as well as supporting the relatively small variability in the ocean exchange.
- The concern about focussing on El Nino and not mentioning La Nina are addressed by including specific references throughout, and by including La Nina anomalies in Figure 9.
- The concern about peak respiration is by-passed by focussing on referring to the $CO_2$ measurements and their susceptibility to dynamical processes. Details of processes influencing other reservoirs in the global carbon budget are beyond our defined scope.

While these additions do not change the basic premise and conclusions of our study, we found the Referee comments extremely useful in communicating our results to a wider community. The referees are now acknowledged.

Thank you for taking on this difficult task

Roger Francey,  4 September 2019

**Response to Referee 1**
The article by Francey et al describes and analyses 25-year composites of interhemispheric (IH) baseline CO2 differences from NOAA, CSIRO and two independent SIO analysis. They show a good agreement between the 4 monitoring networks and explore the influence of atmospheric dynamics on the CO2 IH gradient with a focus on El Niño periods. The results highlight the importance of IH CO2 transfer parametrization in global carbon cycle models. In general, the paper is scientifically sound and worthy of publication; however, the writing needs some modification and improvement. There is also several typos regarding the results hence the paper needs a careful reading/checking.

We have modified, hopefully improved, and more carefully checked, the writing.

After addressing the comments, the manuscript will be suitable for publication.

**General Comments:**

1. Currently the introduction sounds like this paper is submitted to AMT and not ACP, and it does not entirely sound like an introduction. For moments it is too techical and it is hard to understand the aim of the work. I recommend re-framing the introduction, to better attract the readers that are interested about this work. I would suggest to include an overview of the key findings of FF16 and FF18 (relevant to the work in this paper), and higlight better what this work aims to add to the previous findings. Parts of referring to results in FF16 and FF18 are indeed included later in the paper; however, the way/order they are represented is making things hard to link together and to see the big picture (and it makes the paper harder to read). Potentially the advantages/disadvantages in the Introduction could be added into a separate section (or combine with Section 2).

A substantial rewrite of the Introduction is in response to these recommendations. Technical discussion is moved to Section 2 or addressed in an expanded Supplement. The Scientific 'big picture' should be clearer, and the different roles of FF16, FF18 and this paper are more clearly specified. The advantages and disadvantages are important when aiming for internal consistency over three decades. However, because of REF1 comments this discussion is now in the Supplement.

2. Usually Introductions end with a paragraph summarizing the aim of the work. I had the feeling that this comes quite late at the end of Section 2. In addition to re-framing the Introduction I would move the last paragraph from Section 2 to the Introduction. This would also partially solve the missing key findings of FF16 and FF18.

The final paragraph of the introduction (Line 68) now defines the scope of the paper.

3. At the moment the clarification/discussion of few things that can potentially lead to confusion for the readers are missing from the paper. It would be good to just summarize somewhere in the paper: 1) the important ENSO periods discussed in the analysis. The reason for this is that I kept thinking that the highlighted 2010 period in Figure 3 had potentially something to do with the strong 2011 La Niña event (although without reading FF16 first). 2) Why La Niña is not discussed at all and how it would impact IH(e.g., LaNiña periods facilitates interhemispheric mixing of trace gases while ElNiño inhibits interhemispheric exchange... ). In Section4 the authors wrote "Different responses of IH CO2 to wind indices at different ENSO events, and from non-ENSO periods, are discussed in Section 6." and since ENSO includes both El Niño and La Niña, the impact of La Niña should be at least mentioned somewhere. These things are discussed in FF16 and FF18; however, it would be beneficial to add a sentence or two here also to be easier to track/understand the results.

The concerns about La Nina are mentioned in the Abstract " (line 25)", in lines 63-66 of the Introduction and elsewhere the term ENSO is used to include both El Niño and La Niña events. The ONI in strong La Nina years are now included in Figure 9.

4. Page 1 line 24 and elsewhere in the paper: "5-year relatively ENSO-quiet period" I assume the authors meant "5-year relatively El Niño-quiet period" since 2011 was a strong La Niña period, so ENSO-quiet period is misleading.

Agreed, see point 3 for the revised approach

5. Figure 4 - are there emission anomaly uncertainties that could be added to the Figure and included in the discussion in Section 4.1?

To address this point, we have added a 16 DVM model composite, and seasonal Fossil data to Figure 4 to demonstrate their relative magnitudes. Because all surface-to-air fluxes appear very much smaller than required to significantly alter the baseline CO₂ differences, we have not dwelt on obtaining uncertainties from source data.

**Specific Comments:**

\* Section 2 is quite lengthy and could be simplified. A careful reading to condense some of the text would be useful.

A new section 3 is introduced, and the sampling advantages/disadvantages moved to Supplement S3 in order to break up the lengthy section.

\* Page 5 line 183: "e.g. 256 of 300 months have 4 networks" - It seems like the Table shows 268 instead of 256 for mlo-spo, or I misunderstood the Table in which case the authors need to give a better explanation of the Table. Moreover, for mlo-spo with both 268 or 256 the total number of months does not add up to 300. Also regarding the Table, what is the difference between empty boxes and the ones with'-'. If nothing then use consistent marking.

The Table is now updated to 2017 (since SIO spo data became available). The numbers are now consistent, and the main text adjusted accordingly.

\* Page 7 line 228: "It is assumed here that flux variations from ocean sources are much smaller than terrestrial" - it would be good to have some references here. Also just mention why anthropogenic (fossil fuel) emissions are not compared here.

This is now addressed in lines 216, lines 315-320, and Figure 8. The seasonal variation in Southern Ocean fluxes (an important part of the global ocean flux) are similar to terrestrial fluxes from the region but much smaller than NH terrestrial flux variation. High precision continuous $CO_2$ monitoring across the Southern Ocean (Stavert et al., 2019, and personal communication) confirm relatively small variation. Fossil emission seasonal variations are included line 322 and in the new Figure 4(d).

\* Page 9 line 334: It would be good to state why did the authors choose the ONI index as the ENSO index instead of the other indices. It would be also interesting to see if other ENSO indices show the same results (but not strictly necessary to include in this paper if it is too time consuming).

We have re-examined the ONI data and they give essentially the same result as Nino3 and Nin3.4 indices, as stated in line 331.

\*Page9 line337: "there are no significant ONI"-well relative to ElNiño but not relative to La Niña, so the statement is a little bit misleading.

See reworded lines 3333-336.

**Technical Comments:**

\*Page2 1st and 2nd paragraph: It feels like a weird jump between the two paragraphs, a 'linking' sentence between the two would be good.

A linking sentence is provided, lines 33-36.

\* Page 5 line 166, Page 8 line 308, Page 9 line 334: however -> ;however, \* Page 6 line 200: & -> and

Fixed, thanks

\* Page 8 line 271: FF18.These -> missing space before These

Fixed, thanks

\* Figure 5: ITCZ -> Inter Tropical Convergence Zone, the abbreviation was not defined

Fixed, thanks

\* Page 9 line 314: "see Discussion, Section 5" -> see Discussion, Section 6 maybe?

Fixed, thanks

\* Page 9 line 339 ppm.(PgC) -> missing space before (PgC)

Fixed, thanks

\* Page 10 line 258-359: why is this a separate paragraph?

Fixed, thanks

\* northern hemisphere is somewhere capitalized and somewhere not in the text \* NH abbreviation is not defined

Fixed, thanks

\* Figure 3 inconsistency between figure and caption: (a) mlo-cgo, (b) mlo-spo -> (a) mlo-spo, (b) mlo-cgo

Fixed, thanks

\* Figure 7 and 8 please add x axis label

Fixed, thanks

**Response to Referee 2**
**Summary:** Francey et al. present an analysis of monthly data for three measurement stations (MLO, CGO and SPO) to investigate the interhemispheric CO2 difference (IH CO2, defined as concentration difference of MLO-CGO or MLO-SPO) variations over the last 25 years. After comparing the different data sets, the potential biogeochemical drivers for the short-term, seasonal and inter-annual variability of IH CO2 is discussed. The manuscript highlights the ability of different atmospheric transport indices to explain these IH CO2 variations. The manuscript is well-structured and contains important and refined ideas that will be useful in guiding the improvement of GCMs commonly used in global carbon cycle modelling. Its topic is well suited for ACP and surely of interest to the wider scientific community. However, two general comments and a few minor comments should be addressed before publication.

**General comments:** One key argument of this paper is that the variations of IH CO2 cannot be explained by the net uptake of CO2 in the NH terrestrial biosphere. Unfortunately, this assumption is based only on a single DGVM, which are known to be highly uncertain. A comparison of CABLE and other models can be found here: https://journals.ametsoc.org/doi/pdf/10.1175/2008JCLI2378.1. Multiple DGVMs or an ensemble mean of CMIP LSMs would be more robust here.

A 16-DGVM ensemble of extra-tropical NBP from Bastos et al. 2017 is now included (lines222-225), including representation in Figure 4. It confirms the surface flux anomalies are too small to account for IH$\Delta$CO$_2$ changes.

When considering fluxes from optimized emission products (e.g. CarbonTracker-EU) a significantly larger share of IH CO2 could be explained by the NH terrestrial biosphere, although likely not contradicting the finding that IH CO2 is strongly influenced by transport processes.

"We do not discuss air-surface fluxes derived from CO$_2$ data that are less spatially representative, and/or rely on atmospheric transport modelling. The latter introduce additional model degrees of freedom and potentially overestimate terrestrial variability if the variability in atmospheric IH transport is not adequately captured" (lines 224-227).

The manuscript does not use consistent language around the main driver of NH CO2 enhancements. The claim that respiration is maximal in early parts (FEB-APR MAY-JUL) seems unlikely and needs to be supported. However, later the authors refer to NBP and "terrestrial emissions" (which could likely peak in different seasons than autotrophic & heterotrophic respiration. As this is a key issue a clarification would be most helpful.

This is a valid criticism. The scope of this paper is to explore and seek explanation for variations in the baseline CO2 records. While there are clear implications for the global carbon budget, lack of advanced knowledge in every individual component of the budget limits a more comprehensive approach. Reducing uncertainty in the critical atmospheric component of the global budget is the purpose of this study. This is clarified in Lines 68-73 of the rewritten Introduction an in responses to *Specific comment Lines 31, 131, 172, 190*.

**Specific comments:**

Line 31 and Line 44: Why is the proposed method only compared to growth rate studies here and not to 4D-VAR and ensemble kalman filter/smoother data assimilation systems that have been shown to reproduce global scale fluxes with fairly reasonable performance?

The Introduction rewrite aims to clarify this. See also lines 222-225 generated in response to General comments.

Line 55: Figure 1 seems to suggest that with-in network errors do not cancel our. No consistent offset between the different measurement programs was found for all sites. However, to assume that the difference is small compared to the IH CO2 signal seems perfectly reasonable.

We make this assumption.

Line 60: Multi-species observations are mentioned here, but not discussed in this manuscript. Examples of relevant species and isotopes should be given (or paragraph removed).

Multi-species studies, outside the current scope, have the potential to further inform this debate. They are briefly mentioned at line 63 and in the expanded Supplementary information.

Line 63+: The authors do not mention the added value of quasi-continuous data from in-situ observations. If they were available (or used here), issues relating to the impact of different sampling dates and sampling frequencies of the different flask programs could assessed more quantitatively.

The advantages and disadvantages are relevant when aiming for internal consistency over three decades. Challenges to achieving consistent site differences are much greater with the conventional NDIR analysers used over most of this period. This discussion is now focussed in the new Supplement. In relation to spatial differences, NDIR in situ analyses have had calibration limitations due to short lifetime of reference and calibration gases. Flasks analyses in a central laboratory have largely avoided these limitations over decadal timeframes.

Line 122: Unclear what kind of information is available or referenced in Keeling 1998. It would be great to have more details.

It is a brief reflection of the difficulties in maintaining government support for monitoring. It is best articulated in the original publication.

Line 131. Please clarify: the maximum of 7-10ppm of IH $CO_2$ occurs when most exchange occurs between NH and SH? Seems counter-intuitive or do you talk about the specific gradients of two sites here?

Yes, poorly worded. A $CO_2$ partial pressure difference between hemispheres is a prerequisite for net IH exchange. See lines 120-1.

Line 133: Do the offsets really "cancel" here? (see Line 55).

The words used were "largely cancel", which is the case. (It is supported in Figure 4 by the $\Delta IHCO_2$ response to the 1997 GFED anomaly).

Line 155: Why are those definitions of seasons used here and not the more common meteorological or astronomical definitions? Please consider clarifying.

The reasons are stated in lines 141-143: "the particular 3-month seasonal selection distinguishes periods of relatively stable partial pressure differences between hemispheres and the selected seasons also distinguish eddy and mean IH transport mechanisms (FF18)"

Line 167: Why was the decision made not corrected or flag the flask data but "average out" potential outliers by making a composite time series?

This sentence generally describes selection made by each laboratory prior to publication of the monthly averages used here. In the subsequent compositing this data clear statistical outliers are revealed of unknown origin. In the composite these are suppressed by averaging. See lines 115-118.

Line 172: Please provide a reference on the assumption that maximum respiration from NH forests occurs in the seasons in FEB-APR and MAY-JUL? This seems fairly unlikely, as especially FEB-APR is still cold in most of the boreal forest regions in the NH and both autotrophic and heterotrophic respiration typically increase later in the year e.g. in (late) summer. Or please clarify if this refers to net biome productivity or net ecosystem exchange or "terrestrial emissions" (mentioned later in the manuscript).

See response at the end of General Comments above. The wording at line 157 now merely states the widely accepted general reason for the NH $CO_2$ seasonality.

Line 183: The data in table 1 seems inconsistent with the example here. In general, 1992-2016 is mentioned, but later 1992-2017 is shown. (Figure in supplement has 288 months of maximum data, main text figure refers to 300 months maximum).

The Table is now updated to 2017 (since SIO spo data became available). The numbers are now consistent, and the main text adjusted accordingly.

Line 190: Please clarify "generally attributed to NH forest..." this manuscript argues (in later sections) that 1.) the IH CO2 gradient is dominated by transport and that 2.) the variability in the IH CO2 is driven by the MLO time series. Is this consistent?

The seasonality in mlo $CO_2$ is generally attributed to NH forests, and SH seasonality is small by comparison.

However, the anomalies (mean seasonality subtracted) correspond primarily to anomalies in IH transport indices.

Line 199: Why is 2017 now included (see comment L183)
See reply to comment L183.

Line 214: Some technical detail on how the fit was done would be most useful to the reader here, did you perform a sensitivity analysis for different types of splines?
Spline polylines, available in commercial plotting software, link peaks and dips to visually aid discussion.

Line220: Please clarify: the assumption that the global atmospheric CO2 budget is not driven by IH transport but by exchanges to other compartments (biosphere, oceans, etc.) does not have to rely on assuming that transport is correct. The total global atmospheric CO2 budget is the same no matter if CO2 is in the NH or SH. I assume the authors want to argue that the NH versus SH budget might be significantly wrong when not accounting for IH exchange or maybe that the sites used (e.g. MLO) reflect more signals than just emissions and sinks in NH?
Yes. Wording in Section 5 is adjusted to make this clearer.

Line 235+: Why was the CABLE model used. DGVMs are inherently uncertain, so using multiple DGVMs or an ensemble estimate would give a better representation of the range of estimates of NH terrestrial fluxes. Optimized emission products, e.g. CarbonTracker-Europe report significantly higher NH terrestrial uptake (same order of magnitude as IH processes discussed later in the manuscript and in Figure 4). Available for download at: http://www.carbontracker.eu/fluxtimeseries.php [See also general comment]
These issues are addressed in the response to General comments, above.

Line253: The qualitative study on the potential impact of CO2 emissions on global and hemispheric CO2 concentrations seems very instructive. However, a more qualitative estimate using a GCM could be beneficial here.
Maybe, if the GCM has sufficient resolution and upper equatorial tropical parameterisation? It is outside the scope of this study.

Line 286: The authors raise an interesting point here: the MLO-SPO and MLO-CGO data is "effectively identical". Why was SPO data included in this study? A separate study of CGO-SPO could have been more enlightening if it could address the question of CO2 uptake in the Southern Oceans.
This point is specifically addressed in the new introduction. $CO_2$ data quality (measurement and spatial representation) is considered a key issue in this paper, in which the composite plays a central role. The cgo-spo comparison is the best independent evidence for both factors. Note: The Southern Ocean uptake is informed by other records (including in ultra-high precision in situ monitoring at cgo and Macquarie Island plus other Antarctic sites). This is outside the scope of this paper and is the subject of on-going investigation by others (e.g. Stavert et al.). And we speculate that other factors such as seasonality in FF emissions may be involved, see line 315-322.

Line 345: What are the uncertainties of the estimated annual concentration trends and are they statistically significant (and different)?
Statistical uncertainties in trends, are now included throughout the paper. We also provide an estimate of uncertainty in the annual average data of Figure 9, Line 326.

Line 368: Here, the manuscript refers to "terrestrial emissions" is this equal to F(ffco2)+NBP?
Yes, text amended, Line 367.

Line 404: How would the IHCO2 impact the growth rate of CO2 derived at other long term reference sites in the NH. Like Barrow, US and Alert, CA? Could analyzing data from those sites help to further separate the impact of IH mixing versus NH fluxes?
Possibly, but spatial representation and quality of data are limiting factors.

**TRACK CHANGES**

[revised manuscript text omitted]

---

## Author Response (AR2)

Dear Christoph

Thank you and the referees for the opportunity to improve this document. Changes in response to the latest referee comments are detailed below and are identified in the marked-up version.

Best regards

Roger.

REPORT #1
REF#2:
For final publication, the manuscript should be accepted as is
Suggestions for revision or reasons for rejection (will be published if the paper is accepted for final publication)
Specific comments: Line 200: additional space in "peak and t*rough"

**Author response: For consistency replace trough with dip (see marked-up version)**

REPORT#2
REF#1:
Francey et al have acknowledged the proposed changes to their manuscript. My comments have been in most cases sufficiently well responded, and I recommend publication after a very small number of further minor changes: 1. Figure 9 - since the authors completed the comparison, could the results with Nino3 and Nino3.4 also be included somewhere (e.g. in the supplement)? Just stating "(conclusions are similar using Nino3 or Nino3.4)" on line 329 is a little bit vague.  2. In Figure 9 state in figure caption what is the blue dashed line. 3. Line 254 maybe remove the question mark at the end of the sentence? 4. Consistency when writing interhemispheric and inter-hemispheric. There are also other inconsistencies in the writing (e.g., fossil fuel vs Fossil Fuel), hence a detailed re-reading/checking by the authors would be great.

**Author responses:**
- **The Nino3/Nino4 issues is addressed line 321: "The ONI is a 3-month running mean of the Nino3.4 index; similar treatment of Nino3 yields correlation coefficients with Nino3.4 of 0.94 for both the annual maxima and minima from 1992-2018."**
- **The blue dashed line in Figure 9 was a linear regression through the 1992-2016 mlo-spo annual values. Figure 9 is re-drafted, and this linear regression is replaced one through the 1992-2017 mlo-cgo values (orange dashed).**
- **We have taken this opportunity to improve Figure 9 by including error bars on each year for mlo-cgo. The text at line 317 is adjusted accordingly** (in the process correcting a typographical error in the previous estimate of average uncertainty, now 0.08, still much smaller than the observed year-to-year variation).
- **The question mark in "Line 254" has been removed.**
- **The consistency issues for 'inter-hemispheric' an 'Fossil Fuel' are addressed globally by replacing them with 'interhemispheric' and 'fossil fuel'.**
- **In the Figure 3 caption "rectangles" has been replaced by "panels".**

[revised manuscript text omitted]